# Neural activity tracking identity and confidence in social information

Nadescha Trudel[1,2,3]*, Patricia L Lockwood[4,5,6], Matthew FS Rushworth[1,7†], Marco K Wittmann[1,3,8]*†

[1]Wellcome Centre of Integrative Neuroimaging (WIN), Department of Experimental Psychology, University of Oxford, Oxford, United Kingdom; [2]Wellcome Centre for Human Neuroimaging, University College London, London, United Kingdom; [3]Max Planck UCL Centre for Computational Psychiatry and Ageing Research, University College London, London, United Kingdom; [4]Centre for Human Brain Health, School of Psychology, University of Birmingham, Birmingham, United Kingdom; [5]Institute for Mental Health, School of Psychology, University of Birmingham, Birmingham, United Kingdom; [6]Centre for Developmental Science, School of Psychology, University of Birmingham, Birmingham, United Kingdom; [7]Wellcome Centre of Integrative Neuroimaging (WIN), Centre for Functional MRI of the Brain, Nuffield Department of Clinical Neurosciences, John Radcliffe Hospital, University of Oxford, Oxford, United Kingdom; [8]Department of Experimental Psychology, University College London, London, United Kingdom

*For correspondence:
nadeschatrudel@gmail.com (NT);
m.wittmann@ucl.ac.uk (MKW)

†These authors contributed equally to this work

**Abstract** Humans learn about the environment either directly by interacting with it or indirectly by seeking information about it from social sources such as conspecifics. The degree of confidence in the information obtained through either route should determine the impact that it has on adapting and changing behaviour. We examined whether and how behavioural and neural computations differ during non-social learning as opposed to learning from social sources. Trial-wise confidence judgements about non-social and social information sources offered a window into this learning process. Despite matching exactly the statistical features of social and non-social conditions, confidence judgements were more accurate and less changeable when they were made about social as opposed to non-social information sources. In addition to subjective reports of confidence, differences were also apparent in the Bayesian estimates of participants' subjective beliefs. Univariate activity in dorsomedial prefrontal cortex and posterior temporoparietal junction more closely tracked confidence about social as opposed to non-social information sources. In addition, the multivariate patterns of activity in the same areas encoded identities of social information sources compared to non-social information sources.

## Editor's evaluation

The authors used an elegant design to tackle a longstanding question about the extent to which social learning relies on specialized computational and neural mechanisms. They found that learning about ostensible others is more accurate than learning about non-social objects, despite identical statistical information, and that such effects are mediated by brain regions previously implicated in social cognition. These important results should be of interest to a broad range of social, behavioral, clinical, and cognitive neuroscientists.

**eLife digest** People's decisions are influenced by their beliefs, which may be based on advice from other humans or, alternatively, on information from non-human sources such as road signs. But which sources do we find more reliable? Although scientists have studied the importance of social context in the way we process information, it is not fully understood how the brain processes information differently depending on who provides it.

Trudel et al. investigated the differences in the way humans evaluate information from human sources, such as advisors, compared to non-human sources, like inanimate objects, by monitoring brain activity and analyzing the results using a computational approach. In the experiments, 24 participants received a reward for locating a hidden dot on a circle under two different conditions: they either received a clue on the dot's location from an image of a human face (social condition), representing an advisor, or from an inanimate object (non-social condition). Participants received information from many different advisors and inanimate objects, and the accuracy of the clues given by any of them varied from source to source. Each time, participants reported whether they thought the advice was reliable.

The results of monitoring the participants' brain activity showed that they used different strategies when assessing the reliability of advice from a human than when the information came from a non-human source. Additionally, participants based their judgments about an advisor more strongly on past experiences with them, that is, if an advisor had given them good advice in the past, they were more likely to rely on their advice. Conversely, judgments about an inanimate object were based more strongly on recent experiences with that object.

Interestingly, participants were more certain when making judgments about the accuracy of cues given by advisors compared to inanimate objects, and they also updated their assessment of human sources less according to new evidence that contradicted their initial belief. This suggests that people may form more stable opinions about the reliability of sources when they receive information in social contexts, possibly because they expect more consistent behavior from humans. This stability in judgments about advisors was also reflected in the signal of brain areas that are often involved when interacting with others.

The work of Trudel et al. shows that even the suggestion that the source of a piece of advice is human can change how we process the information. This is especially important because humans spend increasingly more time in the digital world. Awareness of our biased assessment of human sources will have implications for designing interactive tools to guide human decision-making as well as strategies to develop critical thinking.

## Introduction

To behave adaptively, humans and many other species learn from their conspecifics which actions to take and which ones to avoid. The human ability to learn not only from direct interactions with the environment, but also from other people contributes critically to our behavioural repertoires as individuals and also to collective culture (*Boyd and Richerson, 2009*; *Gweon, 2021*). When seeking information about which courses of actions to pursue, we routinely turn to others for advice and adjust our own course of action based on the quality of their advice (*Bang et al., 2017*; *Bang et al., 2020*; *Behrens et al., 2008*; *De Martino et al., 2017*; *Hertz et al., 2017*). Clearly, it is important to learn over time whether an advice giver is predicting external events accurately. This process, sampling and learning from other people's advice, is arguably a process that may be very similar to learning from non-social cues about the occurrence of external events (*Akaishi et al., 2016*; *Trudel et al., 2021*). In other words, the cognitive operations underlying this type of information search might occur both in social and in non-social scenarios. Therefore, by comparing information sampling from social versus non-social sources, we address a long-standing question in cognitive neuroscience, the degree to which any neural process is specialized for, or particularly linked to, social as opposed to non-social cognition (*Chang and Dal Monte, 2018*; *Diaconescu al., 2017*; *Frith and Frith, 2010*; *Frith and Frith, 2012*; *Grabenhorst and Schultz, 2021*; *Lockwood et al., 2020a*; *Lockwood et al., 2018*; *Soutschek et al., 2016*; *Wittmann et al., 2018*; *Rushworth et al., 2012*).

Given their similarities, it is expected that both types of learning will depend on common neural mechanisms. However, given the importance and ubiquity of social learning, it may also be that the neural mechanisms that support learning from social advice are at least partially specialized and distinct from those concerned with learning that is guided by non-social sources. However, it is less clear on which level information is processed differently when it has a social or non-social origin. It has recently been argued that differences between social and non-social learning can be investigated on different levels of Marr's information processing theory: differences could emerge at an input level (in terms of the stimuli that might drive social and non-social learning), at an algorithmic level or at a neural implementation level (*Lockwood et al., 2020a*). It might be that, at the algorithmic level, associative learning mechanisms are similar across social and non-social learning (*Heyes, 2012*). Other theories have argued that differences might emerge because goal-directed actions are attributed to social agents which allows for very different inferences to be made about hidden traits or beliefs (*Shafto et al., 2012*). Such inferences might fundamentally alter learning about social agents compared to non-social cues.

There is evidence that at least some parts of the brain are specialized for negotiating social situations (*Noonan et al., 2014*; *Sallet et al., 2011*; *Sliwa and Freiwald, 2017*). For example, neuroimaging studies in macaques show a positive correlation between group size and areas in the temporal lobe (*Sallet et al., 2011*), in an area that has similarities with the temporoparietal junction (TPJ) in humans (*Mars et al., 2013*). In turn, human TPJ and the macaque STS region have been associated and causally linked to the ability to infer other people's beliefs (*Hill et al., 2017*; *Schurz et al., 2017*) and, in monkeys, to inferring the intended actions of others (*Ong et al., 2021*). Dorsomedial prefrontal cortex (dmPFC) is a second important node within this network again both in humans (*Wittmann et al., 2018*; *Hampton et al., 2008*; *Nicolle et al., 2012*; *Piva et al., 2019*; *Suzuki et al., 2012*) and in non-human primates (*Noritake et al., 2018*; *Yoshida et al., 2012*). For example, a recent study demonstrates dmPFC is causally implicated in social cognition and the maintenance of separate representations of oneself and of other individuals even when people interact (*Wittmann et al., 2021*). There are however, studies that report many common brain areas linked to reward and motivational processes in the context of both social and non-social behaviour suggesting at the very least that social information processing is not completely separate (*Behrens et al., 2008*; *Boorman et al., 2013*; *Ruff and Fehr, 2014*; *Will et al., 2017*; *Zink et al., 2008*).For example, tracking the probability of being correct or successful, often referred to as 'confidence' in one's choice, has been shown to correlate with activation in perigenual anterior cingulate cortex in both social (*Wittmann et al., 2016*) and non-social (*Bang and Fleming, 2018*) settings. Here, we seek to understand the behavioural and neural computations that differentiate learning from social and non-social information sources and to further test at which level of information processing these differences occur.

It has been proven difficult to dissociate social as opposed to non-social cognitive functions. One reason is the lack of controlled within-subject experimental designs that have rarely been implemented in previous social neuroscience studies. Another reason might be the type of cognitive process that is studied; only cognitive functions that might plausibly occur in either social and non-social scenarios allow for a fair comparison. Here, we attempted to optimize our experimental design to address both these considerations in order to examine whether there are any fundamental differences in behavioural and neural computations when seeking information from social as opposed to non-social sources. First, we implemented a tightly controlled within-subject design with two experimental sessions that were matched in their statistical properties and only differed in their framing (social vs. non-social) and in their precise visual features. This ensured that differences between conditions were confined to the nature of the information source – social or non-social – and therefore that any differences in behaviour or neural activity could only attributed to the social/non-social difference. Second, we compared learning from other people's advice with learning from non-social cues about the occurrence of external events. On every trial of the experiment, participants witnessed the precision with which a social cue (i.e., advice) or a non-social cue predicted a target. Over time, participants learned about the quality of these predictions and formed estimates of how confident they were in a cue, that is how precisely they expected the target to be predicted by the cue. We show that humans appear more certain in the performance accuracy of social predictors (advisors) compared to non-social predictors. This was evident in more stable confidence judgements across multiple timescales in the social as opposed to the non-social condition; there was a stronger reliance on representations

made in the past and a weaker integration of new contradictory evidence. By using a computational approach, we could associate differences in the stability in confidence about social advisors to differences in the Bayesian estimates of participants' subjective beliefs. We found that two brain areas, dmPFC and posterior TPJ (pTPJ) showed specificity in their processing of social information, not only in their average activation but also by encoding the identity of social cues in their multivariate pattern activation.

## Results

We used a social and non-social version of a previously validated paradigm (*Trudel et al., 2021*) during which participants learned about the performance of social and non-social cues in predicting a target on the circumference of a circle. On every trial, participants received advice on where to find the target (*Figure 1 a, b* , *Figure 1—figure supplement 1*). In the social version, faces represented advisors that predicted a target position (a flower's position on the circumference of a circle) (*Figure 1a*). Some advisors predicted the target accurately and others less accurately. Participants were instructed that advisors had previously been players of a similar task during which time they could directly learn about the target location (*Figure 1c*, *Figure 1—figure supplement 2*). In other words, the other players had learned directly about the target location, while in the main experiment, participants could only infer the target locations from the predictions made by these previous players who now acted as advisors, but not directly from the target. Performance was defined by the size of an angular error, which represented the distance between the location of the predicted flower position (represented by a black dot) and the true position of the flower (represented by a yellow dot).

In the non-social version, participants were instructed to collect fruits (the fruit position was now the target position and again it was represented by a yellow dot) that could either fall a small or large distance from the tree (this was now the predictor and again it was represented by a black dot). Importantly, task versions were completely matched in their experimental design and statistical properties inherent to the predictors and only differed in terms of their framing towards the participants. Thus, if there were differences in behaviour or neural activity between the conditions, then these could not have been attributed to any difference in protocol but only to a predisposition to treat the two types of predictors differently. Note that, the critical predictive stimuli and targets were, respectively, simply yellow and black dots appearing on a similar circle's circumference in both conditions.

On every trial, participants indicated their confidence in the advisor's performance. A symmetrical confidence interval (CI) with a random initial width appeared around the (social or non-social) prediction. The CI could be changed continuously to make it wider or narrower, by pressing buttons repeatedly (one button press resulted in a change of one step in the CI). In this way, participants provided what we refer to as an 'interval setting'. Reward was received when the target fell within the chosen interval and the number of points increased when the interval size was small. A narrow interval could be set when participants selected a predictor with a small angular error and when they were certain about the predictor's angular error. Hence, trial-wise confidence judgements allowed us to inspect participants' beliefs about the performance of the advisor and their uncertainty in this performance estimate. In both social and non-social conditions, the participants' beliefs about the predictor's performance could be updated during the outcome phase when they witnessed the angular error between the predictive black dot and the target yellow dot. Participants were presented with multiple social and non-social predictors across six blocks. In both social and non-social settings, predictors varied in the accuracy with which they predicted the target, that is their average angular error (*Figure 1c*). For each predictor accuracy in the social condition, there was always a predictor in the non-social condition with a matched accuracy level.

### Confidence in the performance of social and non-social advisors

We compared trial-by-trial interval setting in relation to the social and non-social advisors/predictors. When setting the interval, the participant's aim was to minimize it while ensuring it still encompassed the final target position; points were won when it encompassed the target position but were greater when it was narrower. A given participant's interval setting should, therefore, change in proportion to the participant's expectations about the predictor's angular error and their uncertainty about those expectations. Even though, on average, social and non-social sources did not differ in the precision

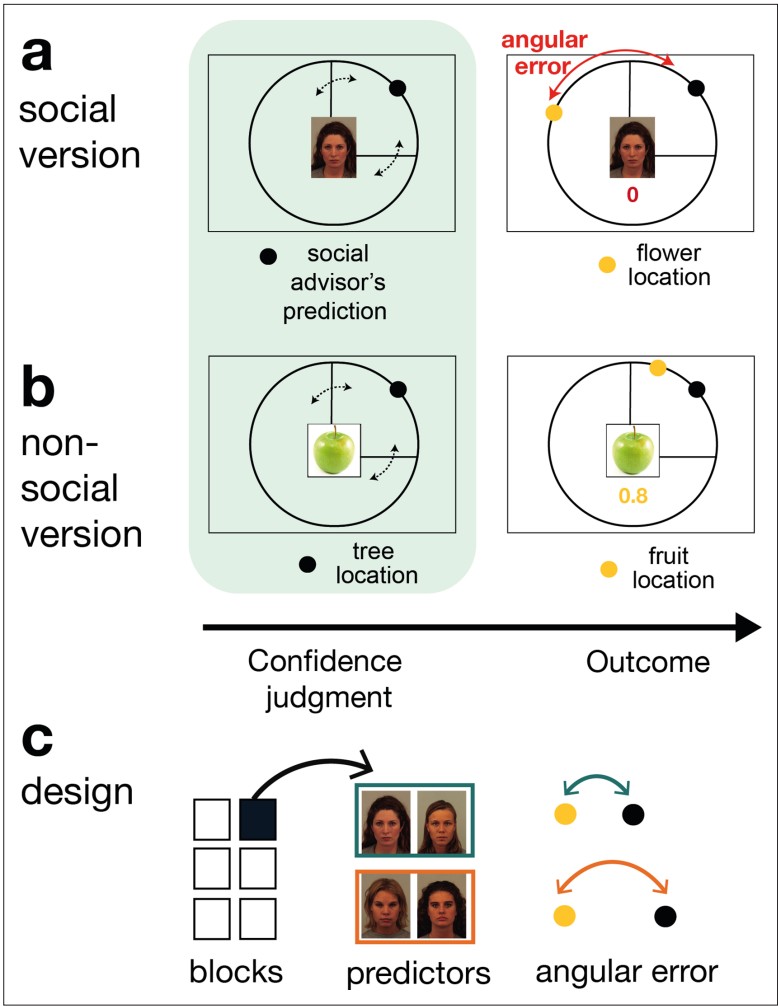

**Figure 1.** Social and non-social task versions and experimental design. Participants performed a social and non-social version of an information-seeking task. Versions only differed in their framing but were matched in their statistical properties. Participants learned about how well predictors (social advisors (**a**) and non-social cues (**b**)) estimated the location of a target on the circumference of a circle. On every trial, participants received advice on where to find the target. Participants indicated their confidence in the likely accuracy of performance by changing the size of a symmetrical interval around the predictor's estimate (black dot, confidence phase): a small interval size indicated that participants expected the target to appear close to the predictor's estimate. During the outcome phase, participants updated their beliefs about the predictor's performance by inspecting the angular error, that is the distance between the predictor's estimate (black dot) and true target location (yellow dot). (**a**) Social version. Information was given by social advisors that were shown as facial stimuli. Participants were instructed that advisors represented previous players that learnt about the target distribution themselves. Crucially, participants could not learn about the target (yellow dot) location themselves and could only infer the target locations from the predictions (i.e., social advice; black dot) from these previous players. (**b**) Non-social version. Participants selected between fruits (yellow dots) that fell with a different distance from their tree location (black dot). The aim was to select fruits with the smallest distance. Again, participants could not learn about the fruit location themselves, but had to infer it from the distance to their trees. (**c**) Design. Each session comprised six experimental blocks (in total 180 trials). Each block included four new predictors (here as an example shown for the social condition), of which there were two good predictors (on average small angular error) and two bad predictors (on average big angular error).

The online version of this article includes the following figure supplement(s) for figure 1:

**Figure supplement 1.** Task design.

**Figure supplement 2.** Social instruction procedure.

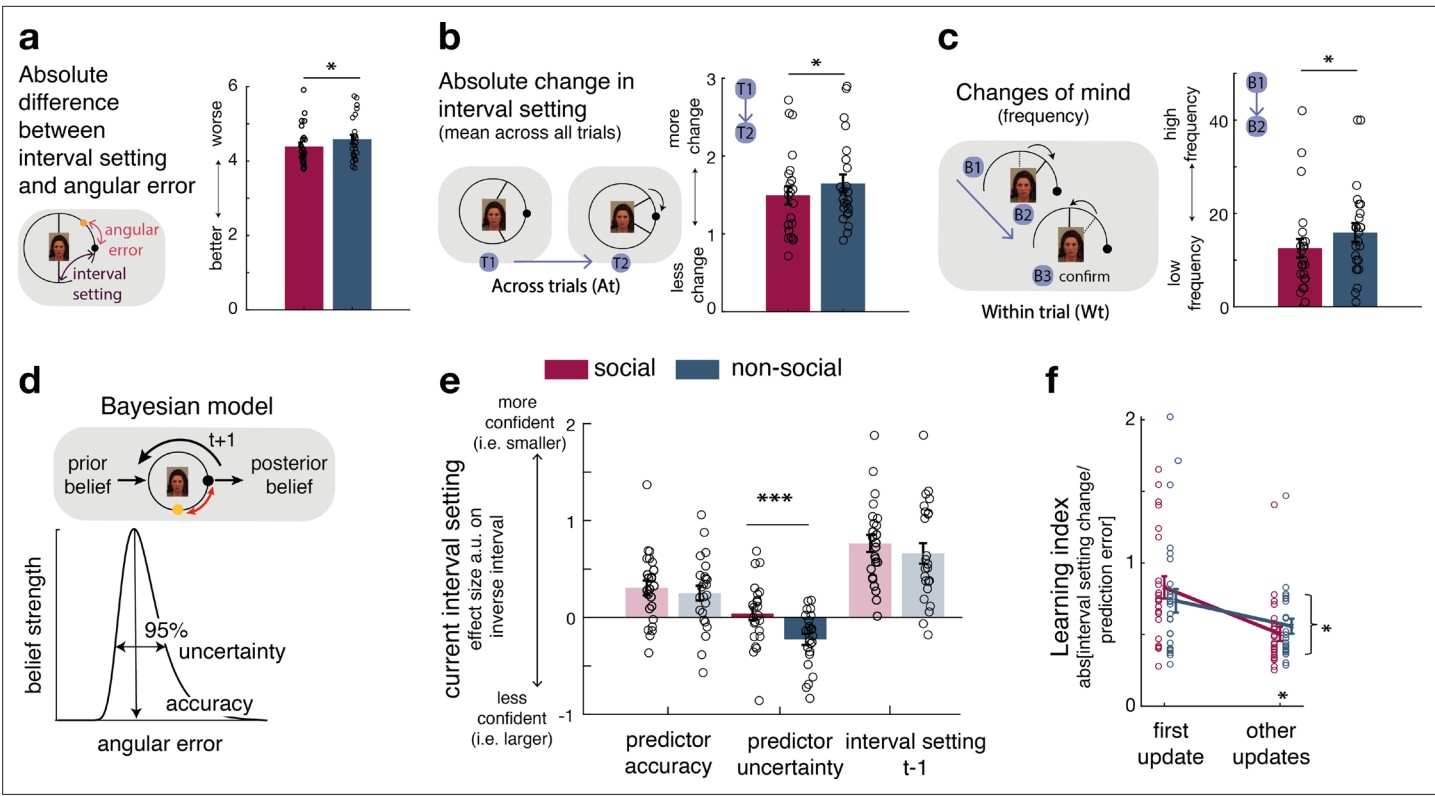

**Figure 2.** Interval setting in relation to the performance of social and non-social advisors. (**a**) Participants' interval setting (corresponded more closely to the true predictor performance error angular error) of the social advisor compared to the non-social cue. Changes of mind in confidence judgements are measured on two timescales: across trials, by comparing the absolute change in interval setting between the current and next judgement of the same predictor (**b**) and within trials, by changes of mind in interval setting just prior to committing to a final judgement (**c**). Within trial changes of mind were defined by a specific sequence of the two last button presses (from button 1 (B1) to button 2 (B2)) before confirming the final judgement (button 3 (B3)): participants first decreased the interval (B1: more confident) and then increased it again (B2: less confident). There were fewer changes of mind about confidence judgements, as indexed by interval setting, both across trials (**b**) and within trials (**c**) for social advisors compared to non-social predictors. Note that panels a–c show raw scores. (**d**) Bayesian model. We used a Bayesian model to derive trial-wise belief distributions about a predictor's performance. The prior distribution updates after observing the angular error of the selected predictor, resulting in a posterior distribution which serves as prior distribution on the next encounter of the same predictor. At each trial, two belief estimates were derived: first, the belief in the accuracy (point-estimate corresponding to the angular error belief that was strongest) and second, the uncertainty in that belief (95% range around the point-estimate) (for details about the Bayesian model, see ***Figure 1—figure supplement 2***). (**e**) Confidence general linear model (GLM): We used a linear regression model to predict trial-wise interval setting. Note that we have sign-reversed the interval setting to make the measure more intuitive: a greater value in a regressor indicates that it is associated with more confidence (i.e., smaller interval setting). Participants were more confident when they selected a predictor they believed to be accurate. In addition, interval settings were impacted by subjective uncertainty in beliefs but only in the case of the non-social predictors: a larger interval was set when participants were more uncertain in their beliefs about the predictor's accuracy. (**f**) Learning index. A trial-wise learning index shows that after the first update, changes in the next confidence judgement, as indexed by the next intervals setting, for social advisors are less strongly impacted by deviations between the current interval setting and angular error (i.e., prediction error) (n = 24, error bars denote standard error of the mean [SEM] across participants, asteriks denote significance level with * p<0.05, *** p<0.005).

The online version of this article includes the following source data and figure supplement(s) for figure 2:

**Source data 1.** Source data include data shown in ***Figure 2a-c***.

**Figure supplement 1.** Additional behavioural results.

**Figure supplement 2.** Task statistics and Bayesian model.

with which they predicted the target (***Figure 2—figure supplement 1***), participants gave interval settings that differed in their relationships to the true performances of the social advisors compared to the non-social predictors. The interval setting was closer to the angular error in the social compared to the non-social sessions (***Figure 2a***, paired *t*-test: social vs. non-social, *t*(23) = −2.57, p = 0.017, 95% CI = [−0.36 −0.4]). Differences in interval setting might be due to generally lower performance in the non-social compared to social condition, or potentially due to fundamentally different learning

processes utilized in either condition. We compared the mean reward amounts obtained by participants in the social and non-social conditions to determine whether there were overall performance differences. There was, however, no difference in the reward received by participants in the two conditions (mean reward: paired *t*-test social vs. non-social, $t(23) = 0.8$, p = 0.4, 95% CI = [−0.007 0.016]), suggesting that interval setting differences might not simply reflect better or worse performance but instead suggest quantitative differences in qualitatively similar learning mechanisms.

Estimating the angular error of an advisor accurately depends on how judgements are adjusted from trial to trial. We compared changes in interval settings on two timescales. First, we did this on a longer timescale, by comparing the absolute change between the current and next judgement of the same predictor (which might only be presented again several trials later, *Figure 2b*). Second, we did this on a within-trial timescale, by comparing changes of mind prior to the final judgement: participants adjusted the CI's size by individual button presses, allowing inspections of changes of mind within a trial according to a specific sequence of button presses prior to the final judgement. Changes of mind within a trial were indicated by first becoming more confident (selecting smaller intervals) and then becoming less confident (selecting a bigger interval) before the final judgement (*Figure 2c*, inset). Participants made fewer adjustments of their confidence judgements across both timescales for social advisors: participants' absolute changes in their confidence judgements from trial to trial were lower (*Figure 2b*, paired *t*-test: social vs. non-social, $t(23) = −2.7$, p = 0.0125, 95% CI = [−0.3 −0.04]) and they exhibited fewer changes of mind before committing to a final interval setting (paired *t*-test: social vs. non-social, $t(23) = −2.4$, p = 0.025, 95% CI = [−6.3 −0.45]). In sum, these results suggest that participants are more certain in their interval setting for social compared to non-social information sources, as they change estimates about an advisor less drastically across trials and commit to their judgement within each trial with less hesitation.

We tested the hypothesis that participants' certainty in their own beliefs influenced their interval setting differently in social and non-social settings. We used a Bayesian model that allowed dissociation of two aspects of subjective beliefs: first, belief in how well the predictor will perform on a given trial (a belief in the predictor's 'accuracy') and second, the participant's uncertainty associated with the predictor's predicted accuracy ('uncertainty' of the belief) (*Figure 2d*, *Figure 2—figure supplement 2*; *Trudel et al., 2021*). Both aspects of a belief might impact on the adjustment of the interval setting. Accuracy is the more likely belief parameter of the two to affect interval setting because accuracy reflects a point-estimate of the most likely angular error that an advisor has shown in the past. However, narrow interval settings, which have the potential to yield more reward if they encompass the target (*Figure 2a*) can only be employed when participants believe the advice given to be a true reflection of the target location (the advice is accurate) and when they are certain in their belief that the advice will be accurate (certainty in the advice's accuracy). We, therefore, tested the impact of both belief parameters on trial-wise interval settings and examined whether their impact differed when making judgements about social or non-social cues. For both social and non-social predictors, participants were more confident and set a smaller interval when they believed the predictor to be accurate in their target prediction (paired *t*-test: social vs. non-social, $t(23) = 0.6$, p = 0.558, 95% CI = [−0.13 0.24]; one sample *t*-test: non-social, $t(23) = 3.2$, p = 0.003, 95% CI = [0.09 0.41]; one sample *t*-test: social, $t(23) = 4$, p < 0.0001, 95% CI = [0.15 0.46], *Figure 2e*, *Figure 1—figure supplement 2*). However, when judging non-social predictors, but not social predictors, interval settings were additionally widened as a function of subjective uncertainty: participants set a larger interval when they were more uncertain about the non-social cue's accuracy to predict the target (paired *t*-test: social vs. non-social, $t(23) = 3.5$, p = 0.0018, 95% CI = [0.11 0.42]; one sample *t*-test: non-social, $t(23) = −3.9$, p < 0.0001, 95% CI = [−0.35 −0.1]; one sample *t*-test: social, $t(23) = 0.6$, p = 0.56, 95% CI = [−0.1 0.19], *Figure 2e*). The difference between social and non-social conditions was replicated when using a mixed-effects model (*Figure 2—figure supplement 1*). Uncertainty about the accuracy of social advisors, however, did not have the same impact: participants' interval settings for the social advisors appeared only to be guided by the peak of the Bayesian probability distribution, as indicated by the accuracy effect.

We have seen that participants have less changeable estimates of the performances of social advisors compared to non-social advisors (*Figure 2b*), and this relative increase in certainty about the value of social advice as opposed to non-social information is also reflected in the Bayesian estimates of participants' subjective beliefs (*Figure 2d, e*). One possibility is that this increased certainty is

caused by relying on a longer-term memory of the observed advice in the social compared to the non-social context. Participants might form an opinion about the performance of social advisors more quickly and are therefore less likely to change it in face of new and possibly contradictory evidence. We therefore tested whether the degree of error-driven behavioural adaptation is different between social and non-social conditions. In other words, we tested whether participants changed their interval setting from one trial to the next to a smaller degree as a function of the discrepancy between their interval setting and the angular error (i.e., prediction error). We calculated a trial-by-trial learning index, similar to a learning rate in a reinforcement learning (RL) model (*Barto, 1998*; *Wittmann et al., 2020*; *Lockwood and Klein-Flügge, 2020b*), that measured the change between the current and next interval setting given the prediction error (*Figure 2f*; Methods, *Equation 2 and 3*). A smaller learning index shows less error-driven adjustment of one's interval setting. Learning indices between conditions were similar for the first observation but declined more steeply for social compared to non-social predictors for the remaining updates (repeated-measures analysis of variance [ANOVA]: interaction between group (social, non-social) × time point (first update, other remaining updates), $F(1,23) = 5.8$, $p = 0.024$, *Figure 2f*). Hence, after the first update, interval setting in the social condition was subsequently less influenced by prediction errors that was the case in the non-social case (paired *t*-test: social vs. non-social for remaining only, $t(23) = 2.7$, $p = 0.01$, 95% CI = $[-0.95\ -0.13]$, *Figure 2f*). As a consequence, judgements relied more on observations made further in the past in the social condition. This is consistent with the observation that participants repeatedly set similar intervals across time in the social condition (i.e., fewer changes across trials) (*Figure 2b*), and that these settings were less impacted by their own subjective uncertainty (*Figure 2e*).

## The impact of noise in belief updating in social and non-social conditions

So far, we have shown that, in comparison to non-social predictors, participants changed their interval settings about social advisors less drastically across time, relied on observations made further in the past, and were less impacted by their subjective uncertainty when they did so (*Figure 2*). In further exploratory analyses, we used Bayesian simulation analyses to investigate whether a common mechanism might underlie these behavioural differences. We tested whether the integration of new evidence differed between social and non-social conditions; for example, *recent* observations might

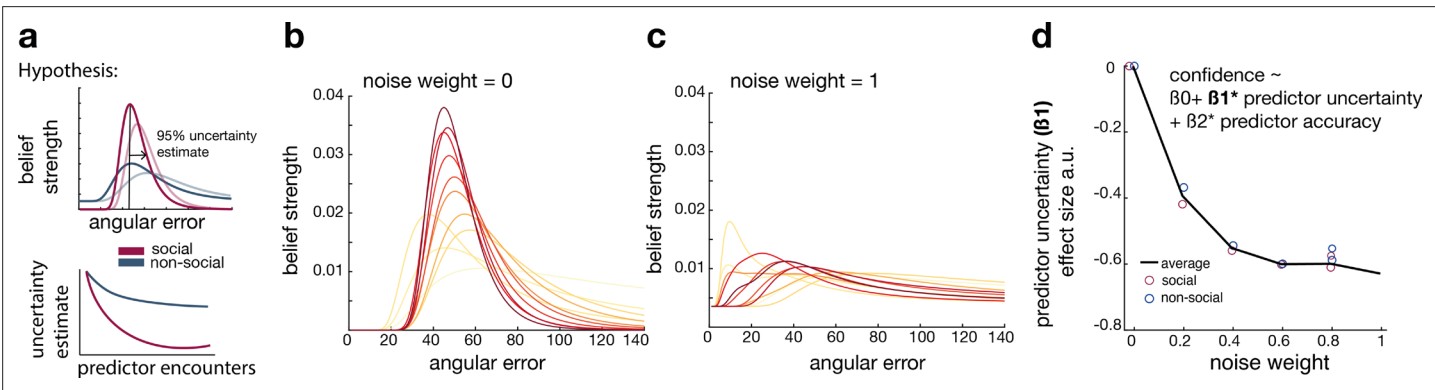

**Figure 3.** Noisy Bayesian model captures differences observed between social and non-social conditions. (**a**) We tested whether there are differences in the belief update made about social or non-social information. We hypothesized belief updates to be inherently more uncertain (i.e., wider distribution) for non-social compared to social information (upper schema) and that this uncertainty decays slower with additional encounters of the same predictor (lower schema). (**b, c**) As in previous work, we manipulated the width of the belief distribution by adding a uniform prior to each belief update. A noise parameter of zero reflects the original Bayesian model, while a noise parameter of one (**c**) reflects a very noisy Bayesian model; we additionally used noise levels on a continuum between 0 and 1 (not shown here). (**d**) Interval setting was simulated by increasing the weighting of a uniform distribution into trial-wise belief updates. We applied a general linear model (GLM) to each noise weight and each condition, here we show the regression coefficient for the 'predictor uncertainty' effect (*y*-axis) on simulated interval setting for a continuum of noise weights (*x*-axis). Increasing the noise weight leads to a more negative 'predictor uncertainty' effect on interval setting. Regression weights are averaged across participants within either the social (red circle) or non-social condition (blue circle); the black line represents the average across conditions.

The online version of this article includes the following figure supplement(s) for figure 3:

**Figure supplement 1.** Comparison between original and noisy Bayesian model.

be weighted more strongly for non-social cues while *past* observations might be weighted more strongly for social cues. Similar ideas were tested in previous studies, when comparing the learning rate (i.e., the speed of learning) in environments of different volatilities (**Behrens et al., 2007**; **Meder et al., 2017**). We tested these ideas using established ways of changing the speed of learning during Bayesian updates (**Allenmark et al., 2018**; **Yu and Cohen, 2008**). We hypothesized that participants reduce their uncertainty quicker when observing social information. Vice versa, we hypothesized a less steep decline of uncertainty when observing non-social information, indicating that new information can be flexibly integrated during the belief update (**Figure 3a**).

We manipulated the amount of uncertainty in the Bayesian model by adding a uniform distribution to each belief update (**Figure 3b, c**; **Equations 10 and 11**). Consequently, the distribution's width increases and is more strongly impacted by recent observations (see example in **Figure 3—figure supplement 1**). We used these modified Bayesian models to simulate trial-wise interval setting for each participant according to the observations they made by selecting a particular advisor in the social condition or other predictor in the non-social condition. We simulated CIs at each trial. We then used these to examine whether an increase in noise led to simulation behaviour that resembled behavioural patterns observed in non-social conditions that were different to behavioural patterns observed in the social condition.

First, we repeated the linear regression analysis and hypothesized that interval settings that were simulated from noisy Bayesian models would also show a greater negative 'predictor uncertainty' effect on interval setting resembling the effect we had observed in the non-social condition (**Figure 2e**). This was indeed the case when using the noisy Bayesian model: irrespective of social or non-social condition, the addition of noise (increasing weight of the uniform distribution to each belief update) led to an increasingly negative 'predictor uncertainty' effect on confidence judgement (new **Figure 3d**). The absence of difference between the social and non-social conditions in the simulations suggests that an increase in the Bayesian noise is sufficient to induce a negative impact of 'predictor uncertainty' on interval setting. Hence, we can conclude that different degrees of noise in the updating process are sufficient to cause differences observed between social and non-social conditions. Next, we used these simulated interval settings and repeated the descriptive behavioural analyses (**Figure 2b, f**). An increase in noise led to greater changes of confidence across time and a higher learning index (**Figure 3—figure supplement 1**). In summary, the Bayesian simulations offer a conceptual explanation that can account for both the differences in learning and the difference in uncertainty processing that exist between social and non-social conditions. The key insight conveyed by the Bayesian simulations is that a wider, more uncertain belief distribution changes more quickly. Correspondingly, in the non-social condition, participants express more uncertainty in their confidence estimate when they set the interval, and they also change their beliefs more quickly. Therefore, noisy Bayesian updating can account for key differences between social and non-social condition.

## dmPFC and pTPJ covary with social confidence and encode social advisor identity

Next, we tested whether there are brain areas that covary with interval setting that are made about social compared to non-social advisors. Behaviourally, we have shown that judgements about social advisors might reflect the presence and influence of relatively constant and unchanging representations in memory. Therefore, a combination of the trial-wise interval setting made in the current (time point $t$) and past trial (time point $t − 1$) for the same predictor was used to test for differences between social and non-social advisor judgements (contrast: current + past interval setting). We sign-reversed the interval setting measure such that a higher index now reflected greater confidence in the selected predictor. We focussed on two a priori independent regions of interest (ROIs), dmPFC (**Wittmann et al., 2016**) and pTPJ (**Mars et al., 2013**) because of their previous association with inference about beliefs held by others (**Schurz et al., 2017**; **Hampton et al., 2008**; **Nicolle et al., 2012**; **Piva et al., 2019**; **Suzuki et al., 2012**; **Wittmann et al., 2016**; **Saxe, 2006**); a process that is crucial to the current task. Hence, we tested whether blood-oxygen-level-dependent (BOLD) signal in these brain areas changed in tandem with variation in the intervals participants set. This parametric contrast thus captures trial-by-trial variation in the confidence that participants have in the predictor and is independent of the mean confidence levels. Note, however, that the mean confidence level was not significantly different between social advisors and non-social predictors (paired *t*-test: mean confidence,

social vs. non-social, *t*(23) = −1.7, p = 0.11, 95% CI = [−0.66 0.07]), even though we observed greater stability of interval setting for social advisors (*Figure 2b*). Activation in both dmPFC and pTPJ covaried with combined confidence judgements significantly more strongly for social compared to non-social advisors (*Figure 4a*, paired *t*-test: social vs. non-social for dmPFC, *t*(23) = −2.5, p = 0.019, 95% CI = [3 31]; *Figure 4b*, paired *t*-test: social vs. non-social for pTPJ: *t*(23) = −2.3, p = 0.03, 95% CI = [1 22.5]). Next, we performed an exploratory whole-brain analysis and found that these activations in dmPFC and pTPJ were also detectable when examining activity across the brain and using standard cluster-correction techniques to adjust for multiple comparisons when making judgements about social advisors (family-wise error (FWE) cluster corrected with *z*-score >2.3 and p < 0.05, *Supplementary file 1*). In addition, activity was significantly related to the same measure in the social condition after whole-brain cluster-correction, in some other areas previously linked with social cognition such as the medial frontal pole and posterior cingulate/medial parietal cortex (*Figure 4c, d*, for a complete list of cluster-corrected brain areas, see *Supplementary file 1*).

Our behavioural results showed that social judgements might be especially impacted by beliefs formed over longer timescales: for example, judgements about social advisors changed less drastically across time, even if new information at the time of the trial outcome was contradictory to one's current belief (*Figure 2f*). We tested whether such a differentiation between current (*t*) and past (*t* − 1) interval setting was also apparent in the neural activation profile of dmPFC and pTPJ. In other words, we tested whether pTPJ and dmPFC displayed an activation profile consistent with a greater influence of interval settings over the longer term, as opposed to on just the current trial, to a greater degree in the social as opposed to the non-social condition. Consistent with the behavioural results, activation within both areas covaried with interval settings made over longer timescales rather than just on the current trial (one sample *t*-test, social: dmPFC, past vs. current: *t*(23) = 3.04, p = 0.0058, 95% CI = [7.6 39.8]; pTPJ, past vs. current, *t*(23) = 2.25, p = 0.034, 95% CI = [1.1 26.6]; *Figure 3e, f*; *Figure 4—figure supplement 1*). There was no difference in the strength of neural activity related to interval setting at the two time scales in the non-social condition (*Figure 4—figure supplement 2*).

So far, we have focussed on how the univariate activity in dmPFC and pTPJ covaried with the participants' estimates of the accuracy of predictors. Next, we examined whether the same brain regions carried additional information about the predictors that is orthogonal to what we have examined so far; we sought evidence as to whether the two areas encoded the identities of the cues. We applied a representational similarity analysis (RSA) to BOLD activity patterns evoked by each face and each fruit presentation at the time of response during the confidence phase. We compared the pattern dissimilarity across voxels (measured by the Euclidean distance) in the same a priori independent ROIs in dmPFC and pTPJ as examined previously. For each condition, the Exemplar Discriminability Index (EDI) was calculated which reflects the average dissimilarity in activity elicited across presentations of the same cue (*Figure 3g*, orange area) compared to the average dissimilarity across presentations of different cues (*Figure 3g*, red area) (*Bang et al., 2020*). Hence, a negative EDI indicates that an area's activity changes less when the same advisor/predictor is presented again than when different advisors/predictors are presented suggesting that it encodes representations of the identity of predictors/advisors. The patterns of activation change in both dmPFC and pTPJ suggested that both areas encoded the identities of social cues more strongly than non-social cues (*Figure 3h*; repeated-measures ANOVA: main effect of condition (social, non-social) in 2 (condition: social, non-social) × 2 (area: pTPJ, dmPFC) ANOVA: *F*(1,21) = 5, p = 0.036). An exploratory whole-brain searchlight analysis was performed to test the specificity of dmPFC and pTPJ encoding the identity of social compared to non-social cues. No other cluster-corrected effects were found (*Figure 4—figure supplement 3*). Additional control analyses show that neural differences between social and non-social conditions are not due to the visually different set of stimuli used in the experiment but are instead representing fundamental differences in processing social compared to non-social information (*Figure 4—figure supplement 4*). In summary, the conjunction, univariate and multivariate analyses demonstrate that dmPFC and pTPJ represent beliefs about social advisors that develop over a longer timescale and encode the identities of the social advisors.

## Discussion

We examined the behavioural and neural computations that differentiate learning from social and non-social information sources. Using a tightly controlled within-subject design in which conditions

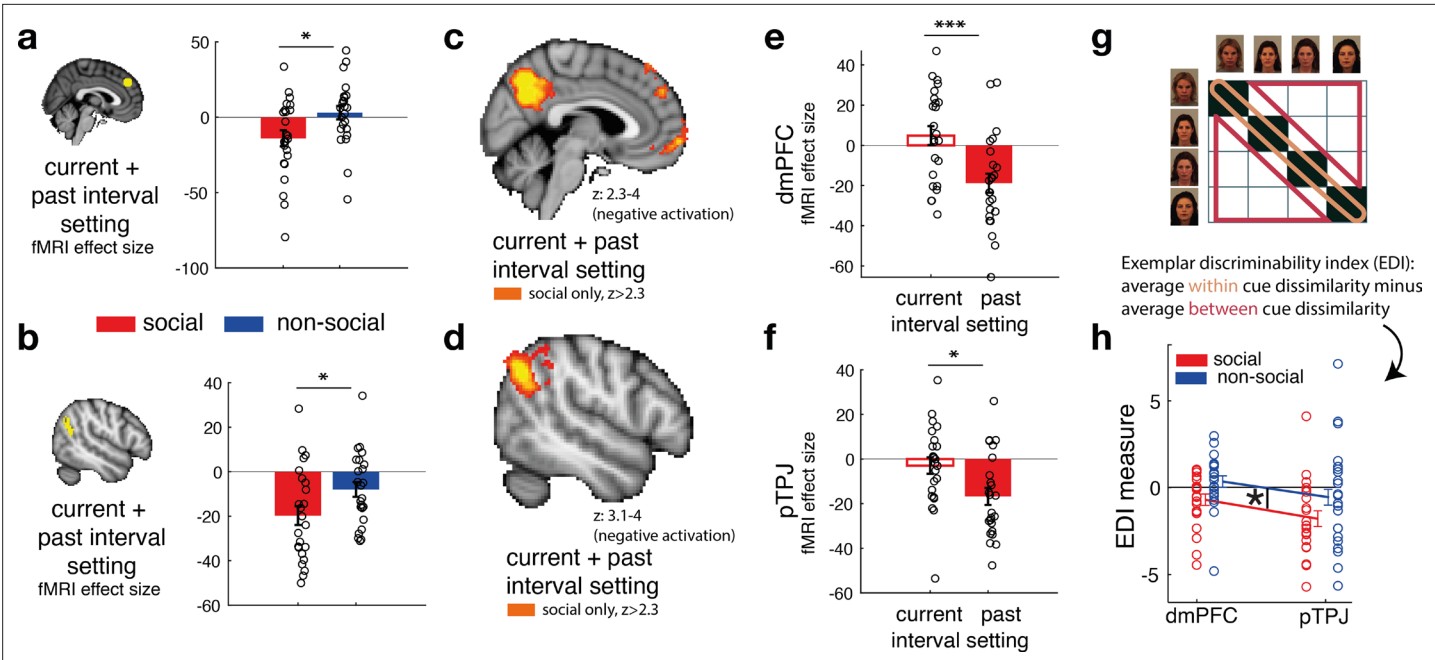

**Figure 4.** Social confidence and advisor identity in dorsomedial prefrontal cortex (dmPFC) and posterior temporoparietal junction (pTPJ). We used a region-of-interest approach based on two independent a priori regions of interest (ROIs) in dmPFC (**a**, left) and pTPJ (**b**, left). Activation in both areas covary with the combination of current (at trial *t*) and past (at trial *t* − 1) judgements for social advisors compared to non-social advisors. We performed a whole-brain exploratory analysis and show cluster-corrected activation for the same contrast as in (**a, b**) for the social condition in dmPFC (**c**) and pTPJ (**d**). Analyses were family-wise error (FWE) cluster corrected with *z*-score >2.3 and p < 0.05, but for visualization purposes, *z*-thresholds differ between both images (dmPFC, *z*: 2.3–4 and pTPJ, *z*: 3.1–4). Note that we use the same colour scheme to indicate social (red colour) and non-social conditions (blue colour) throughout this figure; however, the depicted whole-brain activations are of negative polarity (***Supplementary file 1***). This means that both dmPFC (**e**) and pTPJ (**f**) activities reflect interval setting represent judgements that are informed by past observations compared to current observations (see ***Figure 4—figure supplement 1*** for whole-brain results) in the social condition. Activation in both areas scaled significantly more strongly with past compared to current judgements for social advisors. (**g**) A representational similarity analysis was applied to the blood-oxygen-level-dependent (BOLD) activity patterns evoked by each predictor (faces and fruits in the social and non-social conditions, respectively) at the time of response during the confidence phase. We compared pattern dissimilarity across voxels (measured by the Euclidean distance) between all pairwise combinations of social and non-social cues. Exemplar Discriminability Index (EDI) was calculated to compare the dissimilarity of the same cues (across diagonal, orange area) to the dissimilarity of different cues (off-diagonal, red area). (**h**) A negative EDI indicates multivariate activity patterns are more similar when the same cue, as opposed to different cues are seen. Multivariate patterns within dmPFC and pTPJ encoded the identity of social cues more strongly than it encoded the identities of non-social predictors (n = 24, error bars denote standard error of the mean [SEM] across participants; whole-brain effects are FWE cluster corrected with *z*-score >2.3 and p < 0.05, asteriks denote significance levels with * p<0.05, *** p<0.01).

The online version of this article includes the following source data and figure supplement(s) for figure 4:

**Source data 1.** Source data include data shown in ***Figure 4a*** (contrast: current + past interval setting in dorsomedial prefrontal cortex [dmPFC] region of interest [ROI]), ***Figure 4b*** (contrast: current + past interval setting in posterior temporoparietal junction [pTPJ] ROI), ***Figure 4e*** (separate effects of current and past interval setting in dmPFC ROI for social condition only), ***Figure 4e*** (separate effects of current and past interval setting in pTPJ ROI for social condition only), ***Figure 4h*** (Exemplar Discriminability Index [EDI] values for dmPFC ROI), and ***Figure 4h*** (EDI values for pTPJ ROI).

**Figure supplement 1.** Whole-brain cluster-corrected effects for past and current interval setting in social and non-social conditions.

**Figure supplement 2.** No difference between neural activity related to current and past interval settings for non-social predictors in dorsomedial prefrontal cortex (dmPFC) (panel **a**) and posterior temporoparietal junction (pTPJ) (panel **b**).

**Figure supplement 3.** Exploratory whole-brain searchlight analysis.

**Figure supplement 4.** Control analyses.

only differed in their social or non-social framing, we asked participants to make trial-wise confidence judgements, by setting a visual interval, about the advice of social and non-social information sources. Ideally, confidence should increase with the accuracy of one's choice (***Bang and Fleming, 2018***; ***Kiani and Shadlen, 2009***; ***Lak et al., 2020***). Here, participants did not match their confidence to the likely accuracy of their own performance, but instead to the performance of another social or non-social advisor. Participants used different strategies when setting intervals to express their confidence in the

performances of social advisors as opposed to non-social advisors. A possible explanation might be that participants have a better insight into the abilities of social cues – typically other agents – than non-social cues – typically inanimate objects. We used a computational approach to test the interplay of two belief estimates that impact interval setting: the belief in how accurate the advice/prediction is and the subjective uncertainty in that belief. Intervals were impacted by several factors; however, it was the influence of subjective uncertainty about the performance estimates that distinguished social and non-social interval setting. Participants selected a greater interval size when making their confidence judgements on non-social trials, indicating that they were more uncertain about the performance accuracies of the non-social cues compared to the social cues.

Being less influenced by ones' uncertainty about the value of social advice might help explain why people sometimes fail to use evidence to update their assessments of social advisors (*Fleming et al., 2018*; *Rollwage et al., 2020*). Detecting uncertainty in environmental contingencies or about one's own beliefs is achieved with fine-tuned metacognitive abilities (*Miyamoto et al., 2021*) and is essential for behavioural adaptation (*Trudel et al., 2021*; *Badre et al., 2012*). Participants were less influenced by their subjective uncertainty when estimating the performances of social advisors. Moreover, they changed and updated their beliefs about social cues less than about non-social cues when they observed the outcomes of their predictions. Repeating a particular interval size instead of adjusting it from trial to trial might be a better strategy in the current experiment because all advisors had, on average, a stable performance level. However, opting for such a strategy only in the context of social advisors might reflect a natural tendency to expect other social agents to behave consistently and exhibit stable performance levels as soon as an initial learning period is passed. It may be because we make just such assumptions that past observations are used to predict performance levels that people are likely to exhibit next (*Wittmann et al., 2021*; *Wittmann et al., 2016*). An alternative explanation might be that participants experience a steeper decline of subjective uncertainty in their beliefs about the accuracy of social advice, resulting in a narrower prior distribution, during the next encounter with the same advisor. During a series of exploratory analyses, we used Bayesian simulations to investigate how uncertainty about beliefs changed from trial to trial and showed that belief updates about non-social cues were consistent with a noisier update process that diminished the impact of experiences over the longer term. From a Bayesian perspective, greater certainty about the value of advice means that contradictory evidence will need to be stronger to alter one's beliefs. In the absence of such evidence, a Bayesian agent is more likely to repeat previous judgements. Just as in a confirmation bias (*Kappes et al., 2020*), such a perspective suggests that once we are more certain about others' features, for example, their character traits, we are less likely to change our opinions about them.

One potential explanation for the assumption of stable performance for social but not non-social predictors might be that participants attribute intentions and motivations to social agents. Even if the social and non-social evidence are the same, the belief that a social actor might have a goal may affect the inferences made from the same piece of information (*Shafto et al., 2012*). Social advisors first learnt about the target's distribution and accordingly gave advice on where to find the target. If the social agents are credited with goal-directed behaviour then it might be assumed that the goals remain relatively constant; this might lead participants to assume stability in the performances of social advisors. However, such goal-directed intentions might not be attributed to non-social cues, thereby making judgements inherently more uncertain and changeable across time. Such an account, focussing on differences in attribution in social settings aligns with a recent suggestion that any attempt to identify similarities or differences between social and non-social processes can occur at any one of a number of the levels in Marr's information theory (*Lockwood et al., 2020a*). Here, we found that the same algorithm was able to explain social and non-social learning (a qualitatively similar computational model could explain both). However, the extent to which the algorithm was recruited when learning about social compared to non-social information differed. We observed a greater impact of uncertainty on judgements about social compared to non-social information. We have shown evidence for a degree of specialization when assessing social advisors as opposed to non-social cues. At the neural level we focussed on two brain areas, dmPFC and pTP. These brain areas carry signals associated with belief inferences about others, for example how confident we think other people are about their own choices (*Bang et al., 2022*). Further, recent combined fMRI–TMS studies have also demonstrated the causal importance of these activity patterns for social inference (*Hill et al., 2017*; *Wittmann et al., 2021*). Similar processes might be at play in the current study during

which participants make inferences about the performance of the social advisor in locating the target. We examined dmPFC and pTPJ, using ROIs derived from previous studies, and showed that these areas represent a combination of present and past confidence judgements that are made about social advisors and that they do this more strongly for social than for non-social cues.

A long-standing question in cognitive neuroscience is the degree to which any neural process is specialized for, or particularly linked to, social as opposed to non-social cognition. The current results suggest that dmPFC and pTPJ activity patterns are different during social cognition and that they carry information that is related to the performance differences that are seen at a behavioural level during a social as opposed to non-social task. Interestingly, it seems that activation in dmPFC and pTPJ was particularly prominent when representing judgements from previous encounters compared to the present judgement. This was apparent not only in univariate activity in these areas that covaried with confidence in advice but also in the multivariate activity patterns that encoded the identity of social agents; RSA showed that activation patterns in both areas encoded the identity of social cues better than the identity of non-social cues. The absence of similar differences in visual cortex suggests that the effects cannot be attributed to basic visual differences between the stimuli used in the two tasks. Hence, pTPJ and dmPFC seem particularly linked to the navigation of social interactions.

## Methods
### Participants
Thirty participants took part in the experiment of which some were excluded because they fell asleep during the scan ($N$ = 2), exhibited excessive head motions ($N$ = 1) or because they prematurely terminated the sessions ($N$ = 3). The final sample consisted of 24 participants (14 female, age range 19–35). Each participant took part in two versions of the experiment: a social and non-social version. The order of versions was counterbalanced across participants. No statistical method was used to pre-determine the sample size. However, the sample size used is larger than those reported in previous publications (*Boorman et al., 2011*). Moreover, we are using a within-subject design to increase statistical power to detect a true difference between conditions. The study was approved by the Central Research Ethics Committee (MSD-IDREC-C1-2013-13) at the University of Oxford. All participants gave informed consent.

### Experimental procedure
All participants performed two sessions on separate days that only differed in their framing of the experimental task (social or non-social). Each session consisted of 1 hr of magnetic resonance imaging (MRI) scanning and 30 min of instruction including practice trials outside the scanner. Participants learnt the trial procedure and response mapping during practice trials. Practice trials were independent from the main task; they included different stimuli and data and thereby guaranteed that participants encountered predictors for the first time during the scanning session. Participants received £15 per hour and an additional bonus depending on task performance (per session: £5-£7). After both sessions were completed, participants completed a debriefing questionnaire probing their understanding of the task. We used data previously reported by *Trudel et al., 2021* but now with a fundamentally different aim. Previously, Trudel et al. had focussed on uncertainty guided exploration of reward-predicting cues regardless of whether the cues were social or non-social in nature. Accordingly, their analyses collapsed across data obtained in the social and non-social conditions. Now, the question of exploration was no longer of primary importance but instead the focus was on the impact of social versus non-social cue differences on behaviour and neural activity. Here, we investigated behaviour and neural activity in the second phase of each trial during which participants reported their confidence judgements. By contrast, *Trudel et al., 2021* focussed on behaviour and neural activity in the first phase of each trial, the decision phase (*Figure 1—figure supplement 1*).

### Framing of social and non-social task versions
Each participant performed a social and non-social version of the task (*Figure 1*). Tasks were matched in their statistical properties, and only differed in the stimuli (predictors that were used to indicate an information source). In the social task, the predictors were represented with facial stimuli. They represented advisors who predicted a target position (a flower's position on the circumference of a

circle represented by a yellow dot). Advisors differed in in how well they predicted the target location and the participant's aim was to select advisors who made target predictions that were closest to the subsequent actual target positions; if participants followed the guidance of more accurate predictors then they were more likely to win more points (see Experimental design) which ultimately led to a higher payout. In the non-social version, participants were instructed that they were to collect fruits that again lay on the circumference of a circle and which were again represented by yellow dots. Different fruits fell with different distances from their tree and the aim was to select the fruits that fell the smallest distance from their trees. Thus, the fruit trees served an analogous role to the faces in the social condition. In both cases, the stimuli were predictors that estimated the subsequent position of yellow dotes on the circumference of a circle. Just as the advisors in the social condition varied in the accuracy with which they made their predictions about the subsequent yellow dot position, the fruit trees also varied in the accuracy with which they made their predictions about the subsequent yellow dot position. Crucially, the statistical relationships between predictors and predicted items were varied in identical ways across both the social and non-social stimuli. In both conditions, there were some more accurate and less accurate predictors (discussed more in Experimental design). Thus, if there were differences in behaviour or neural activity between the condition then this could not have been attributed to any difference in protocol but only to a predisposition to treat the two types of predictors differently. Stimuli for both task versions were drawn from a database and randomized across participants.

During the experimental task, participants could not learn about the target location directly, but instead had to make a binary selection between predictors during the first phase of the trial, the decision phase (*Figure 1—figure supplement 1*). Having selected a predictor, participants received information on where to find the target in the second phase of the trial (confidence phase). In the social version, participants were instructed that the predictors (advisors in this case) represented previous players who had already performed a behavioural task during which they had had the opportunity to learn about the distribution of the target's location (*Figure 1—figure supplement 2*). During this prior behavioural pre-task, participants were told that the other players had learnt to predict the location of the target through trial and error across time. In other words, during the pre-task, the other players had learned directly about the target location, while in the main experiment, participants could only infer the target locations from the predictions made by these previous players who now acted advisors, but not directly from the target. Analogously in the non-social version of the task, participants were told that different fruits were dispersed more or less closely to their trees, so the tree position was a similar more or less accurate predictor of the target position. The deviation between the predictor's target prediction and the true target position defined the angular error. It was those angular errors that were presented to participants in the main experiment and that enabled participants, albeit indirectly, to estimate the target location. Therefore, in the main experiment, participants were incentivized to try and identify predictors that were more accurate in their predictions and participants were able to do this by observing the angular errors associated with their predictions (*Figure 1a*). This constituted the only way for the participant to estimate the target's location in the main experiment.

## Experimental design

In the experimental sessions, the participants' aim was to select predictors that best predicted the target's location on the circumference of the circle. Participants could learn about a predictor's performance by selecting it in the first phase of the trial (decision phase) and observing the difference between the target prediction and the subsequent true target location (outcome phase) – we refer to this as the angular error (*Figure 1* and *Figure 1—figure supplement 1*). In the second phase of the trial, participants adjusted the size of a symmetrical interval, on a computer monitor, around the predictor's prediction. The interval could be changed by individual button presses that either increased (to indicate less confidence) or decreased (to indicate more confidence) the interval. A third button was used to confirm the final judgement. The interval changed with a precision of 20 steps on each side of the predictor's estimate. The aim of the participants was to select a predictor that predicted the target well. This in turn allowed participants to select a narrow CI around the prediction. When this CI included the target (as revealed in the subsequent outcome phase), then this resulted in a positive reward outcome. The positive outcome was higher if the CI was set very narrowly, and the target still fell within it. If the target fell inside the interval, the payoff magnitude was determined by

subtracting the interval size from 1. If the target fell outside the interval, then participants received a null payoff. Therefore, the payoff ranged between 0 and 1 as follows:

$$\text{Payoff} = \begin{cases} (1 - \text{interval size}) & \text{if target is included} \\ 0 & \text{if target is excluded} \end{cases} \tag{1}$$

Participants transitioned through six blocks, during which they encountered four predictors each depicted by new stimuli in each block. The session comprised two blocks of 15, 30, and 45 trials each. Therefore, both social and non-social sessions comprised 180 trials each. The number of trials remaining was explicitly cued on every trial in the task (*Figure 1—figure supplement 1*). In each part of the experiment (social and non-social), two predictors were associated with a better performance. They had on average a smaller angular error compared to the other two predictors in the block. The predictors' estimates were determined by a Gaussian distribution centred on the true target location but with different standard deviations (standard deviation was either 50 or 70 for the better and worse predictors, respectively). A smaller standard deviation resulted in smaller angular errors. Hence, averaging across the angular errors they had observed allowed participants to learn about the standard deviation associated with a predictor's true performance distribution. For more details, see *Figure 2—figure supplement 1*.

## Bayesian model

We used a Bayesian model (*Trudel et al., 2021 Figure 2—figure supplement 2*) to guide our analyses. We focussed, first, on how participants' behaviour might differ between the social and non-social situations and, second, on how participants estimated their uncertainty about the predictions which was revealed by their setting of the CI. We used a Bayesian model to estimate participants' beliefs about the performance of a social and non-social advisor/predictor. The model was fitted to each social and non-social advisor/predictor to derive belief estimates about the distribution that determined the observed angular error, that is the distance between the social and non-social target estimate and the true target location. Angular errors were drawn from a distribution that varied according to the value of sigma ($\sigma$) which refers to the standard deviation of the normal distribution. Across trials, participants observe how well each predictor performs and thereby participants form a belief about the average angular error, that is the estimated sigma (referring to sigma-hat ($\hat{\sigma}$), reflecting an estimate rather than the true value of sigma) of the distribution.

According to Bayes' rule, a belief is updated by multiplication of a prior belief and a likelihood distribution resulting in a posterior belief, that is the belief update (*Figure 2d*). We assumed a uniform prior across parameter space of sigma-hat. In other words, every possible sigma value could underlie the predictor's distribution with equal probability.

$$p\left(\hat{\sigma}\right) = U\left(1, 140\right) \tag{2}$$

The likelihood function assumed a normal distribution of the observed angular error ($x$), and described the probability of the observation $x$ given each possible sigma value:

$$p(x|\hat{\sigma}) = N(x|\mu = 0, \hat{\sigma}) \tag{3}$$

With Bayes rule, a trial-wise posterior distribution proportional to the multiplication of the likelihood and prior distribution:

$$p(\hat{\sigma}|x) \propto p(x|\hat{\sigma})\,p(\hat{\sigma}) \tag{4}$$

where $p(\hat{\sigma})$ is the prior distribution, $p(x|\hat{\sigma})$ is the likelihood function, and $p(\hat{\sigma}|x)$ is the posterior pdf across parameter space. The posterior pdf is the updated belief across sigma space and is used as prior for the next trial of the same predictor.

Each posterior was normalized to ensure that probabilities across all sigma values added up to one:

$$p(\hat{\sigma}|x) = \frac{p(\hat{\sigma}|x)}{\sum p(\hat{\sigma}|x)} \tag{5}$$

We submitted the computational variables from our model to a general linear model (GLM) analysis to predict participants' interval setting. For every trial, we extracted two model-based subjective belief parameters that captured two key features of each participant's likely beliefs about a predictor's performance. First, the belief of how accurately a predictor would predict the target was captured by the mode of the probability density function on each trial associated with the selected predictor:

$$\text{accuracy} = \max\left[p(\hat{\sigma})\right] * (-1) \tag{6}$$

Note that a higher value of $\max\left[p(\hat{\sigma})\right]$ corresponds to a larger angular error; in other words, it corresponds to a larger deviation between the true target location and the predicted target location. To derive an intuitive interpretation, the sign of $\max\left[p(\hat{\sigma})\right]$ was reversed such that now a greater positive value can be interpreted as higher accuracy. In sum, the accuracy estimates as defined in *Equation 5* represents a point-estimate of a belief distribution in sigma-hat ($\hat{\sigma}$).

A second model parameter was derived to reflect a participant's subjective uncertainty in that accuracy belief. On each trial, a percentage (2.5%) of the lower and upper tail of the prior distribution around the estimated sigma value was derived. The difference between sigma values at each of the tails was subtracted to derive the estimated 'uncertainty' variable:

$$\hat{\sigma}_{\text{high}} \leftarrow \text{cumulative}\left(p(\hat{\sigma})\right) = 97.5\%$$
$$\hat{\sigma}_{\text{low}} \leftarrow \text{cumulative}\left(p(\hat{\sigma})\right) = 2.5\% \tag{7}$$
$$\text{uncertainty} = \hat{\sigma}_{\text{high}} - \hat{\sigma}_{\text{low}}$$

Please refer to *Figure 2—figure supplement 2* for a schematic illustration of the Bayesian model.

## Behavioural analyses

To test for differences between social and non-social conditions, we first compared how participants set their interval during the confidence phase. We tested whether there were differences in how they learnt about a social versus a non-social predictor's performance. We compared their interval setting with the true angular error. We computed an index of each participant's behaviour defined as the absolute difference between the true angular error and the size set for the interval that was set. We averaged the resulting indices over sessions and compared the indices in the social and non-social sessions. Next, we tested how participants adjusted their interval settings across two timescales. First, we tested interval settings across trials by calculating – for the same predictor – the absolute difference between the current and next size of the interval setting averaged across all trials within social and non-social conditions (*Figure 2b*). Next, we exploited the fact that the interval setting was modified by individual button presses, which allowed testing for 'changes of mind' within a trial. Changes of mind were defined by a specific button sequence that first decreased (more confident) the interval and then increased (less confident) the interval prior to committing to the final judgement. We identified the number of trials per session that were characterized by such a 'decrease – increase – commit' sequence and compared their frequency between social and non-social conditions (*Figure 2c*).

Next, we used a computational approach to predict interval setting across conditions. We applied a GLM to predict changes in interval settings as a function of model-based and model-free regressors. Note that setting smaller interval sizes indicated higher confidence in an estimate about predictor/advice accuracy. To make this measure more intuitive, we sign-reversed it so that a index value now represented greater confidence in the selected predictor/advisor. Regressors were normalized across all trials (mean of zero and standard deviation of one). The GLM applied to the confidence phase comprised the following regressors and relevant effects are shown in *Figure 3a* (whole GLM is shown in *Figure 2—figure supplement 1*): predictor accuracy, predictor uncertainty, payoff $t-1$ (payoff observed on the last trial ($t$) for the same predictor, confidence judgements on the past three consecutive trials ($t-1$, $t-2$, $t-3$) for the same predictor and the initial starting position of the CI were added (the latter represents a confound regressor).

Next, we tested the degree to which participants changed their interval setting from one trial to the next according to the difference between the current interval that the participant had set when making their estimation of target position and the actual angular error that was subsequently

observed. This difference can be thought of as the participant's prediction error in this task. We calculated a learning index that resembles a learning rate in an RL model, capturing the rate at which evidence observed at the time of the prediction error is integrated into the next estimate. Note that RL models are often fitted to the data to derive a trial-by-trial value estimate, to predict choice behaviour and neural correlates although direct observation of the model parameters is not possible. However here we are provided, by the participants, with explicit indicators, in the form of interval settings, of their confidence in the predictors/advisors, allowing us to derive a learning rate index for every trial. We derived a trial-by-trial learning index by dividing the absolute change in CI that participants set from one trial, $t$, to the next, $t + 1$, by the unsigned prediction error (i.e., absolute difference between the CI on trial $t$ and the angular error [AE] on trial $t$):

$$\text{learning index} = \frac{|CI(t) - CI(t + 1)|}{|CI(t) - AE(t)|} \tag{8}$$

The magnitude of the learning index can be interpreted as follows: zero indicates that participants did not change their interval from one trial to the next. 1 indicates a interval setting change from one trial to the next that that is equivalent to the distance that their CI was away from the target position in the last trial, which would be beneficial if the angular error for the predictor stayed the same across trials. Values bigger than 1 indicate over-adjustment: a change in interval setting that is even bigger than the prediction error the participant had observed (this would happen if participants were expecting the angular error to be even greater on the next trial). Values smaller than 1 indicate under adjustment of the interval setting; participants are updating their intervals by less than the prediction error that they observed.

$$\text{Learning index} = \begin{cases} 0, & \text{nochange} \\ 1, & \text{change} = PE \\ < 1, & \text{change} < PE \\ > 1, & \text{change} > PE \end{cases} \tag{9}$$

## Extension of Bayesian model with varying amounts of noise

We modified the original Bayesian model (*Figure 2d*, *Figure 2—figure supplement 2*) to test whether the integration of new evidence differed between social and non-social conditions; for example, recent observations might be weighted more strongly for non-social cues while past observations might be weighted more strongly for social cues. We tested these ideas using established ways of changing the speed of learning during Bayesian updates (*Allenmark et al., 2018*; *Yu and Cohen, 2008*). For example, an established way of changing the speed of learning in a Bayesian model is to introduce noise during the update process (*Allenmark et al., 2018*; *Yu and Cohen, 2008*). This noise is equivalent to adding in some of the initial prior distribution and this will make the Bayesian updates more flexible to adapt to changing environments. We added a uniform distribution to each belief update; we refer to this as noise addition to the Bayesian model. We varied the amount of noise between $\delta = [0, 1]$, while $\delta = 0$ equals the original Bayesian model and $\delta = 1$ represents a very noisy Bayesian model. The uniform distribution was selected to match the first prior belief before any observation was made (*equation 2*). This $\delta$ range resulted in a continuous increase of subjective uncertainty around the belief about the angular error (*Figure 3b, c*). The modified posterior distribution denoted as $p'(\hat{\sigma}|x)$ was derived at each trial as follows:

$$p'(\hat{\sigma}|x) = p(\hat{\sigma}|x) + (U(1, 140) * \delta) \tag{10}$$

The noisy posterior distribution as $p'(\hat{\sigma}|x)$ was normalized as follows:

$$p'(\hat{\sigma}|x) = \frac{p'(\hat{\sigma}|x)}{\sum p'(\hat{\sigma}|x)} \tag{11}$$

We applied each noisy Bayesian model to participants' interval settings in the social and non-social conditions. Then, we used the model to simulate interval setting at each trial with varying noise

levels and then examined whether an increase in noise results in simulation behaviour that resembles behavioural patterns observed in the non-social conditions that were different to behavioural patterns observed in the social condition. At each trial, we used the accuracy estimate (*equation 6*) to derive a CI associated with the selected predictor. We transformed the accuracy estimate – which represents a subjective belief about a single angular error – into an interval setting by multiplying the trial-specific angular error with the step size. The step size was derived by dividing the circle size by the maximum number of possible steps. Here is an example of transforming an accuracy estimate into an interval: let us assume the belief about the angular error at the current trial is 50 (Methods, *equation 6*). Now, we are trying to transform this number into an interval for the current predictor on a given trial. To obtain the size of one interval step, the circle size (360 degrees) is divided by the maximum number of interval steps (40 steps; note, 20 steps on each side), which results in nine degrees that represents the size of one interval step. Next, the accuracy estimate in radians (0.87) is multiplied by the step size in radians (0.1571) resulting in an interval of 0.137 radians or 7.85 degrees. The final interval size would be 7.85.

We repeated behavioural analyses (*Figure 2b, e, f*) to test whether CIs derived from more noisy Bayesian models mimic behavioural patterns observed in the non-social condition: greater changes of confidence across trials (*Figure 3—figure supplement 1*), a greater negative 'predictor uncertainty' on confidence judgement (*Figure 3—figure supplement 1*), and a greater learning index (*Figure 3d*).

## Imaging data acquisition and pre-processing

Imaging data were acquired with a Siemens Prisma 3T MRI using multiband T2*-weighted echo planar imaging sequence with acceleration factor of two and a 32-channel head-coil (*Trudel et al., 2021*). We acquired slices with an oblique angle of 30 degrees from posterior to anterior commissure to reduce frontal pole signal dropout. The acquisition parameters were: 2.4 × 2.4 × 2.4 mm voxel size, echo time (TE) = 20 ms, repetition time (TR) = 1.03 ms. A field map was acquired for each session to reduce distortions and bias correction was applied directly to the scan. The structural scan had a slice thickness of 1 mm, TR = 1.9 ms, TE = 3.97 ms, and a voxel size of 1 × 1 × 1 mm.

We used FMRIB Software Library (FSL) to analyse imaging data (*Smith et al., 2004*). Pre-processing stages included motion correction, spatial distortion correction by applying the field map, brain extraction, high-pass filtering, and spatial smoothing using full-width at half-maximum of 5 mm. Images were first registered to each participant's high-resolution structural image and then non-linearly registered to the Montreal Neurological Institute (MNI) template using 12 degrees of freedom.

## ROI approach

We used an ROI approach to test differences between social and non-social conditions in two regions. We extracted independent ROIs from dmPFC and pTPJ. dmPFC ROI was calculated with a radius of three voxels centred on MNI coordinate [*x/y/z*: 2,44,36] (*Wittmann et al., 2016*). pTPJ ROI consisted of a 70% probability anatomical mask including both left and right hemispheres. We derived the left pTPJ ROI by warping the mirror image of the right pTPJ (*Mars et al., 2013*) to the left hemisphere. The selected ROIs were transformed from MNI space to subject space and relevant COPEs (contrast of parameter estimates) were extracted for each participant's session (*Figure 4a, b, e, f*).

## Exploratory MRI whole-brain analysis

We used a single GLM to analyse the data. All phases (decision, confidence, and feedback phase) were included into the GLM. The decision phase was time locked to the onset of the predictor display. The confidence phase was time locked to the interval setting response made in the confidence phase. The feedback phase was time locked to the onset of outcome. The duration between trials was drawn from a Poisson distribution with the range of 4–10 s and a mean of 4.5 s. To decorrelate variables of interest between trial phases, short intervals were included between trials (intertrial intervals) and randomly, but equally allocated to either the transition between decision- and confidence phase or confidence- and outcome phase. Each phase was modelled as a constant. We included additional normalized parametric regressors (mean of zero and standard deviation of one) that were modelled as stick functions (i.e., duration of zero). The following parametric regressors were included:

## Decision phase

- chosen–unchosen uncertainty,
- chosen–unchosen accuracy.

## Confidence phase

- current confidence judgement (i.e., the interval size selected on the current trial),
- past confidence judgement (i.e., the interval size selected on the previous encounter of the same predictor),
- number of button responses (that were made to reach the desired interval size; this regressor acted as confound regressor).

## Feedback phase

- payoff.

We calculated the following contrasts for the confidence phase: (current + past confidence judgement) and (current − past confidence judgement). In addition, we included one regressor that was time locked to all button responses. Note that we only analysed the trials that provided both current and past judgements for the same predictor. The results were submitted to two second-level analyses applied separately to social and non-social conditions using FLAME1 (*Figure 4—figure supplement 1*). All results were FWE cluster corrected at p < 0.05 using a cluster-defining threshold of $z > 2.3$.

### Representational similarity analysis

We were interested in the representation of the identity of the predictors. To establish whether neural activity carried information about the identity of predictors, we compared the neural activity that we recorded when the same predictors were presented and compared this to situations in which different predictors were presented. In both cases, the analyses were conducted within our ROIs. To do this, we set up a new whole-brain first-level analysis from which we extracted the resulting COPE maps for each session for each ROI. We used the same approach as in the exploratory MRI whole-brain analysis. In other words, we used the same set of regressors and time locking for the decision and feedback phase as described above ('Exploratory MRI whole-brain analysis'). However, we changed the way of modelling the confidence phase in order to implement an RSA: Because we reasoned that identity representations should be constructed and strengthened with repeated exposure to the same predictor, the trial bundles were constructed from the end to the beginning of the block. For example, the last three occasions on which a predictor was picked comprised one bundle, the preceding three trials with the same predictor comprised the next bundle and so on. Predictor selections from the very beginning of a block that did not make a bundle of three trials were excluded from the analysis. All trial bundles were submitted to a first-level whole-brain analysis with each bundle represented by a separate regressor. Each regressor consisted of three entries time locked to the response within the confidence phase as this was the phase of interest for detecting differences in neural representations of social and non-social cues. In other words, each bundle was time locked to the response made in the confidence phase for the particular trials that were included into the bundle. Duration was set to zero. No other regressors – parametric or non-parametric – were time locked to these events for the confidence phase. We extracted activation from the same independent dmPFC and bilateral pTPJ ROIs as described above and derived the Euclidean distance for all pairwise combinations of z-maps. We calculated the EDI by subtracting the average distances between z-maps of the same predictors from the average distances between bundles of different predictors (while taking care not to include the distance between two instances of precisely the same bundle; *Figure 4e*). A negative EDI is expected when representations of the same cues are less dissimilar than representations of different cues (*Figure 4g*). Two participants were excluded as they represented outliers for both brain regions (dmPFC and pTPJ) and for both session (social and non-social). We determined outliers according to three times of the median absolute deviation (matlab function: 'isoutlier'):

$$\text{Median absolute deviation} = \text{median}(|A_i - \text{median}(A)|) \tag{12}$$

with *A* representing a variable vector and *i* = 1, 2, …, *N*.

## Acknowledgements

We would like to thank Laurence T Hunt and Geoffrey Bird for helpful comments on task design and analysis. We would like to thank Miriam C Klein-Flügge, Kentaro Miyamoto, Lisa Spiering, and Sankalp Garud for helpful comments on earlier versions of the manuscript. The study was funded by a DTC ESRC studentship (ES/J500112/1) to NT and by a Wellcome Trust grant (WT100973AIA) to MFSR.

## Additional information

### Funding

| Funder | Grant reference number | Author |
| --- | --- | --- |
| Economic and Social Research Council | ES/J500112/1 | Nadescha Trudel |
| Wellcome Trust | 221794/Z/20/Z | Matthew FS Rushworth |
| Medical Research Council | Fellowship MR/P014097/1 | Patricia L Lockwood |
| Medical Research Council | Fellowship MR/P014097/2 | Patricia L Lockwood |
| Sir Henry Dale Fellowship | 223264/Z/21/Z | Patricia L Lockwood |
| Jacobs Foundation | Research Fellowship | Patricia L Lockwood |

The funders had no role in study design, data collection, and interpretation, or the decision to submit the work for publication. For the purpose of Open Access, the authors have applied a CC BY public copyright license to any Author Accepted Manuscript version arising from this submission.

### Author contributions

Nadescha Trudel, Conceptualization, Data curation, Formal analysis, Funding acquisition, Validation, Investigation, Visualization, Methodology, Writing – original draft, Project administration, Writing – review and editing; Patricia L Lockwood, Resources, Software, Methodology, Writing – review and editing; Matthew FS Rushworth, Conceptualization, Supervision, Funding acquisition, Methodology, Writing – review and editing, Investigation, Resources; Marco K Wittmann, Conceptualization, Supervision, Methodology, Writing – review and editing, Resources

### Author ORCIDs

Nadescha Trudel http://orcid.org/0000-0002-9372-3640

### Ethics

The study was approved by the Central Research Ethics Committee (MSD-IDREC-C1-2013-13) at the University of Oxford. All participants gave informed consent.

### Decision letter and Author response

Decision letter https://doi.org/10.7554/eLife.71315.sa1
Author response https://doi.org/10.7554/eLife.71315.sa2

## Additional files

### Supplementary files

• Supplementary file 1. Univariate whole-brain results. Relates to *Figure 4* and *Figure 4—figure supplement 1*. Z-values and coordinates for exploratory fMRI-GLM1 for social and non-social conditions (family-wise error [FWE] cluster corrected with $z > 2.3$, $p < 0.05$).

• Transparent reporting form

## Data availability

We have deposited the choice raw data used for the analyses in the OSF repository at https://osf.io/wjuze/. We also provide fMRI beta coefficients for the contrasts shown in Figure 4. Source data is provided with this paper. The above OSF repository includes code that recreates figures depicted in the main text. For the full Bayesian pipeline, which was used to derive regressors for the behavioural GLM depicted in Figure 2e, please refer to the following OSF repository: https://doi.org/10.17605/OSF.IO/D5QZW.

The following dataset was generated:

| Author(s) | Year | Dataset title | Dataset URL | Database and Identifier |
|---|---|---|---|---|
| Trudel N, Lockwood P, Rushworth M, Wittmann M | 2023 | Neural activity tracking identity and confidence in social information | https://doi.org/10.17605/OSF.IO/D5QZW | Open Science Framework, 10.17605/OSF.IO/D5QZW |

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
