## [Editor Report]

The authors used an elegant design to tackle a longstanding question about the extent to which social learning relies on specialized computational and neural mechanisms. They found that learning about ostensible others is more accurate than learning about non-social objects, despite identical statistical information, and that such effects are mediated by brain regions previously implicated in social cognition. These important results should be of interest to a broad range of social, behavioral, clinical, and cognitive neuroscientists.

---

## [Decision Letter]

**Decision letter after peer review:**

Thank you for submitting your article "Neural activity tracking identity and confidence in social information" for consideration by *eLife*. Your article has been reviewed by 2 peer reviewers, and the evaluation has been overseen by a Reviewing Editor and Michael Frank as the Senior Editor. The reviewers have opted to remain anonymous.

Essential revisions:

1) It is not just the "framing" of information as social that differs across conditions, but also the stimuli (i.e., faces vs. fruit). Ideally, the behavioral experiment should be performed again, with the same cover-story, but using abstract shapes or symbols as stimuli, to rule out the possibility that the complexity/familiarity/salience of faces elicit attentional processes that account for the results. If additional data collection is not possible, the authors need to make a compelling case for why the results can be interpreted despite this confound.

2) Make sure to place the main question (i.e., whether humans learn differently from social vs. non-social information) in the context of relevant previous neural and computational work, both in the Introduction and Discussion, and identify what specific novel conceptual insights can be gleaned from these particular results (see the reviewers' comments below for relevant papers).

3) Use model simulations to test some of the explanations proposed in the Discussion (e.g., does changing the learning index of the Bayesian model produce the observed confidence intervals?).

4) Please perform an MVPA searchlight to assess the specificity of the effects in dmPFC and TPJ.

5) Behavioural indices and analyses need to be better justified (e.g., why should the interval size exactly match the angular error?).

6) Please provide a full description of the computational model in the main text.

7) Carefully consider and respond to all the reviewers' comments, appended below. In your reply, list each specific reviewer comment, followed by the response and the relevant revision/rebuttal.

*Reviewer #1 (Recommendations for the authors):*

The authors should provide the full description of the computational model (rather than the schematic illustration).

For the GLM in Figure 2e, I would suggest the authors use the mixed-effect model.

*Reviewer #2 (Recommendations for the authors):*

Regarding point 1 in the public review: I would be happy if the authors simply engaged more deeply with prior theoretical accounts, both in the introduction and in the discussion. However, I do think the theoretical contributions of this paper could be strengthened by building on some of the proposals in the discussion. In the discussion, the authors propose a few reasons why participants' estimates may have been less influenced by subjective uncertainty in the social condition. In particular, participants could have stronger prior expectations about the stability of social sources, or they could show a steeper decline in uncertainty over time. (Though this latter proposal seems contradicted by the results – wouldn't you then expect to see a larger "learning index" after the first update?) The Bayesian model used in this paper could be used to demonstrate the plausibility of these proposals – if you simulate the model forward using different priors or learning rates, does the model capture qualitative patterns in human behavior?

Regarding point 2: It would be informative to pair the multivariate analysis with a whole-brain searchlight, to test whether any other regions show this effect.

[Editors’ note: further revisions were suggested prior to acceptance, as described below.]

Thank you for resubmitting your work entitled "Neural activity tracking identity and confidence in social information" for further consideration by *eLife*. Your revised article has been evaluated by Michael Frank (Senior Editor), Mimi Liljeholm (Reviewing Editor), and two expert reviewers.

The manuscript has been greatly improved and the only remaining request is that you characterize the additional analyses, prompted by the first round of reviews, as exploratory, since they were not planned in the original submission.

*Reviewer #1 (Recommendations for the authors):*

The authors have adequately addressed all the concerns.

*Reviewer #2 (Recommendations for the authors):*

I thank the authors for their thoughtful and thorough responses to reviewer comments. My past review raised two key concerns. First, I suggested that the paper could engage more deeply with past work on the computational basis of social learning, in order to ground their discussion on the differences between social and non-social learning. Second, I raised the concern that condition differences between social and non-social stimuli could be driven by lower-level features, such as the attentional salience or visual distinctiveness of the stimuli. Overall, I believe that the authors have gone above and beyond to address these concerns.

First, the authors revised their introduction and discussion to cite past scholarship in social learning. But, beyond that, they also substantially expanded their analyses to disentangle differences between the underlying mechanisms driving behavior in social and non-social conditions. The authors found that participants did not have different prior expectations about the performance of social vs. non-social predictors. Instead, using a modified, noisy Bayesian model, the authors suggest that differences between conditions may be driven by degrees of uncertainty in the belief update and that greater noise in the update process leads to larger changes in confidence across different encounters with the predictor. I agree that these analyses have significantly improved the manuscript and made a new theoretical contribution in their own right, as they suggest a common computational mechanism underlying observed behavioral differences between conditions.

If I can offer a small suggestion: Because these analyses and hypotheses were not planned in the initial submission, it would be helpful to specify that they are exploratory/an extension when they are first presented in the main text (p. 10-11).

Second, the authors added several control analyses as a supplement. In order to test whether condition differences could be accounted for by attentional engagement, the authors inspected the constant of the GLM. They found no condition differences unaccounted for by the parametric regressors included in the GLM, either in attentional regions or in the regions that were the focus of their study. Next, in order to test whether social and non-social stimuli differed in their visual distinctiveness, the authors used RSA to compare the EDI between conditions in early visual areas. Again, the authors found no difference, suggesting that observed condition differences are unlikely to be driven by these lower-level features.

---

## [Author Response]

Reviewer #1 (Recommendations for the authors):The authors should provide the full description of the computational model (rather than the schematic illustration).

We have now integrated a full description of the Bayesian model into the Methods section.

Bayesian model. We used a Bayesian model ^18^ (Figure 2 —figure supplement 2) to guide our analyses. We focussed, first, on how participants’ behaviour might differ between the social and non-social situations and, second, on how participants estimated their uncertainty about the predictions which was revealed by their setting of the confidence interval. We used a Bayesian model to estimate participants’ beliefs about the performance of a social and non-social advisor/predictor. The model was fitted to each social and non-social advisor/predictor to derive belief estimates about the distribution that determined the observed angular error, i.e. the distance between the social and non-social target estimate and the true target location. Angular errors were drawn from a distribution that varied according to the value of sigma (σ) which refers to the standard deviation of the normal distribution. Across trials, participants observe how well each predictor performs and thereby participants form a belief about the average angular error, i.e. the estimated σ (referring to sigma-hat (σ^), reflecting an estimate rather than the true value of sigma) of the distribution.

According to Bayes’ rule, a belief is updated by multiplication of a prior belief and a likelihood distribution resulting in a posterior belief, i.e. the belief update (Figure 2d). We assumed a uniform prior across parameter space of σ-hat. In other words, every possible σ value could underlie the predictor’s distribution with equal probability.

p(σ^)=U(1,140)

(2)

The likelihood function assumed a normal distribution of the observed angular error *(x)*, and described the probability of the observation *x* given each possible sigma value:

p(x|σ^)=N(x|μ=0,σ^).

(3)

With Bayes rule, a trial wise posterior distribution proportional to the multiplication of the likelihood and prior distribution:

p(σ^|x)p(x|σ^)p(σ^)

(4)

where p (σ^) is the prior distribution, p(x | σ^) is the likelihood function and p(σ^| x), is the posterior pdf across parameter space. The posterior pdf is the updated belief across σ space and is used as prior for the next trial of the same predictor.

Each posterior was normalised to ensure that probabilities across all σ values added up to one:

p(σ^|x)=pσ^(x)Σp(σ^x)

(5)

We submitted the computational variables from our model to a general linear model (GLM) analysis to predict participants’ interval setting. For every trial, we extracted two model based subjective belief parameters that captured two key features of each participant’s likely beliefs about a predictor’s performance. First, the belief of how accurately a predictor would predict the target was captured by the mode of the probability density function on each trial associated with the selected predictor:

accuracy = max [p (σ^)] * (-1)

(6)

Note that a higher value of max [p (σ^)] corresponds to a larger angular error; in other words, it corresponds to a larger deviation between the true target location and the predicted target location. To derive an intuitive interpretation, the sign of max [p (σ^)] was reversed such that now a greater positive value can be interpreted as higher accuracy. In sum, the accuracy estimates as defined in equation (5) represents a point-estimate of a belief distribution in σ-hat (σ^).

A second model parameter was derived to reflect a participant’s subjective uncertainty in that accuracy belief. On each trial, a percentage (2.5%) of the lower and upper tail of the prior distribution around the estimated σ value was derived. The difference between σ values at each of the tails was subtracted to derive the estimated ‘uncertainty’ variable:

σ^_high ⟵_ cumulative (p (σ^)) = 97.5%

σ^_low ⟵_ cumulative (p (σ^)) = 2.5% uncertainty = σ^_high_ – σ^_low_

(7)

Please refer to the Figure 2 —figure supplement 2 for a schematic illustration of the Bayesian model.

For the GLM in Figure 2e, I would suggest the authors use the mixed-effect model.

Thank you for this suggestion. We agree that a mixed-effect model is, indeed, a sensitive approach to compare the differences between social and non-social conditions as they take the data dependence into account^19^. In the behavioural GLM (Figure 2e), we showed that there are differences in the impact of Bayesian uncertainty estimates on trial-wise confidence judgments about a predictor’s performance: participants were less impacted by the uncertainty around the accuracy estimate when making confidence judgments about social advisors compared to non-social cues. Hence, the aim of the mixed-effect model is mainly to test whether such a difference in the ‘predictor uncertainty’ effect can be replicated. We used a random slope mixed-effect model (lme function in matlab).

The random slope mixed-effect model of interest includes all regressors of interest as fixed effects (Figure 2e and Figure 1 —figure supplement 2), random slopes for each fixed effect as well as a random intercept. We additionally included a binary variable (condition_ID) to denote the social (condition=1) and non-social (condition=0) condition. All variables were normalised. The effect of interest is the interaction effect between the condition variable and the fixed effect ‘predictor uncertainty’.

To evaluate the random slope mixed-effect model, we compared two models – one model that included all regressors of interest [H-alternative] to a model that included all regressors *except the effect of interest ‘predictor uncertainty’ effect and the interaction ‘predictor uncertainty x groupID’* [H-null]. Then, we can compare the model fit and evaluate whether the addition of uncertainty variables is meaningful. Noteworthy, when doing such a random slope mixed-effect model comparison, it is crucial to keep the ‘predictor uncertainty’ variable as random slope in both models. In the syntax of the lme function, the models were specified as follows:

H-null:

Confidence ~ (predictor accuracy + payoff_t-1 + confidence_t-1)* conditionID + ((predictor accuracy + predictor uncertainty + payoff_t-1 + confidence_t-1)*conditionID | subjects).

H-alternative:

Confidence ~ (predictor accuracy + *predictor uncertainty* + payoff_t-1 + confidence_t-1)* conditionID + ((predictor accuracy + predictor uncertainty + payoff_t-1 + confidence_t1)*conditionID | subjects).

First, a likelihood ratio test was used to compare the goodness of fit of the two competing models. The alternative model that included the additional ‘predictor uncertainty’ variable as fixed effect and its interaction with the conditionID was a better model of the data (*χ*2(2) = 22.238, p=.483e-05). Next, we tested whether there was a significant interaction of ‘uncertainty predictor x conditionID. When using a random slope mixed-effect model, we replicated the difference between conditions (‘uncertainty predictor x conditionID, estimate = -0.178, SE = 0.07, p = 0.017) – again showing a difference in the impact of subjective uncertainty when making confidence judgments about social advisors or non-social cues.

We included the results in the main text and Supplementary material as follows:

Main text, page 9:

“However, when judging non-social predictors, but not social predictors, interval settings were additionally widened as a function of subjective uncertainty: participants set a larger interval when they were more uncertain about the non-social cue’s accuracy to predict the target (paired t-test: social vs. non-social, t(23) = 3.5, p=0.0018, 95% CI = [0.11 0.42]; one sample t-test: non-social, t(23) = -3.9, p<0.0001, 95% CI = [-0.35 -0.1]; one sample t-test: social, t(23)=0.6, p=0.56, 95% CI = [-0.1 0.19], Figure 2e). The difference between social and non-social conditions was replicated when using a random slope mixed effect model (Figure 2 —figure supplement 1, legend). Uncertainty about the accuracy of social advisors, however, did not have the same impact: participants’ interval settings for the social advisors appeared only to be guided by the peak of the Bayesian probability distribution, as indicated by the accuracy effect.”

Reviewer #2 (Recommendations for the authors):Regarding point 1 in the public review: I would be happy if the authors simply engaged more deeply with prior theoretical accounts, both in the introduction and in the discussion. However, I do think the theoretical contributions of this paper could be strengthened by building on some of the proposals in the discussion. In the discussion, the authors propose a few reasons why participants' estimates may have been less influenced by subjective uncertainty in the social condition. In particular, participants could have stronger prior expectations about the stability of social sources, or they could show a steeper decline in uncertainty over time. (Though this latter proposal seems contradicted by the results – wouldn't you then expect to see a larger "learning index" after the first update?) The Bayesian model used in this paper could be used to demonstrate the plausibility of these proposals – if you simulate the model forward using different priors or learning rates, does the model capture qualitative patterns in human behavior?

We thank the reviewer for this suggestion. In response, apart from engaging more deeply with prior theoretical accounts, we have undertaken further analyses that aim to disentangle the underlying mechanisms that might differ between both social and nonsocial conditions might provide additional theoretical contributions. We followed the advice of the review and show additional model simulations and analyses that aim to disentangle the differences in more detail. These new results allowed clearer interpretations to be made.

In the current study, we show that judgments made about non-social predictors were changed more strongly as a function of the subjective uncertainty: participants set a larger interval, indicating lower confidence, when they were more uncertain about the non-social cue’s accuracy to predict the target. In response to the reviewer’s comments, the new analyses were aimed at understanding under which conditions such a negative uncertainty effect might emerge.

Prior expectations of performance

Related to the first point of the reviewer, we compared whether participants had different prior expectations in the social condition compared to the non-social condition. One way to compare prior expectations is by comparing the first interval set for each advisor/predictor. In such a way, we test whether the prior beliefs before observing any social or non-social information differ between conditions. Even though this does not test the impact of prior expectations on subsequent belief updates, it does test whether participants have generally different expectations about the performance of social advisors or non-social predictors. There was no difference in this measure between social or non-social cues (Author response image 1; paired t-test social vs. non-social, t(23) = 0.01, p=0.98, 95% CI = [-0.067 0.68]).

**Author response image 1. sa2fig1:** Confidence interval for the first encounter of each predictor in social and non-social conditions. There was no initial bias in predicting the performance of social or non-social predictors.

Learning across time

We have now seen that participants do not have an initial bias when predicting performances in social or non-social conditions. This suggests that differences between conditions might emerge across time when encountering predictors multiple times. We tested whether inherent differences in how beliefs are updated according to new observations might result in different impacts of uncertainty on interval setting between social and non-social conditions. More specifically, we tested whether the integration of new evidence differed between social and non-social conditions; for example, *recent* observations might be weighted more strongly for non-social cues while *past* observations might be weighted more strongly for social cues. This approach was inspired by the reviewer’s comments about potential differences in the speed of learning as well as the reduction of uncertainty with increasing predictor encounters. Similar ideas were tested in previous studies, when comparing the learning rate (i.e. the speed of learning) in environments of different volatilities^12,13^. In these studies, a smaller learning rate was prevalent in stable environments during which reward rates change slower over time, while higher learning rates often reflect learning in volatile environments so that recent observations have a stronger impact on behaviour. Even though most studies derived these learning rates with reinforcement learning models, similar ideas can be translated into a Bayesian model. For example, an established way of changing the speed of learning in a Bayesian model is to introduce noise during the update process^14^. This noise is equivalent to adding in some of the initial prior distribution and this will make the Bayesian updates more flexible to adapt to changing environments. It will widen the belief distribution and thereby make it more uncertain. Recent information has more weight on the belief update within a Bayesian model when beliefs are uncertain. This increases the speed of learning. In other words, a wide distribution (after adding noise) allows for quick integration of new information. On the contrary, a narrow distribution does not integrate new observations as strongly and instead relies more heavily on previous information; this corresponds to a small learning rate. So, we would expect a steep decline of uncertainty to be related to a smaller learning index while a slower decline of uncertainty is related to a larger learning index. We hypothesized that participants reduce their uncertainty quicker when observing social information, thereby anchoring more strongly on previous beliefs instead of integrating new observations flexibly. Vice versa, we hypothesized a less steep decline of uncertainty when observing non-social information, indicating that new information can be flexibly integrated during the belief update (new Figure 3a).

We modified the original Bayesian model (Figure 2d, Figure 2 —figure supplement 2) by adding a uniform distribution (equivalent to our prior distribution) to each belief update – we refer to this as noise addition to the Bayesian model^14,21^. We varied the amount of noise between δ = [0,1], while δ = 0 equals the original Bayesian model and δ = 1 represents a very noisy Bayesian model. The uniform distribution was selected to match the first prior belief before any observation was made (equation 2). This δ range resulted in a continuous increase of subjective uncertainty around the belief about the angular error (Figure 3b-c). The modified posterior distribution denoted as *p*′(σ^| x) was derived at each trial as follows:

*p*′(σ^| x) = *p* (σ^| x) + (*U*(1,140) ∗ δ)

(10)

The noisy posterior distribution as p′(σ^|x) was normalised as follows:

p′(σ^|x)=p′(σ^|x)∑p′(σ^|x)

(11)

We applied each noisy Bayesian model to participants’ choices within the social and nonsocial condition.

The addition of a uniform distribution changed two key features of the belief distribution: first, the width of the distribution remains larger with additional observations, thereby making it possible to integrate new observations more flexibly. To show this more clearly, we extracted the model-derived uncertainty estimate across multiple encounters of the same predictor for the original model and the fully noisy Bayesian model (Figure 3 —figure supplement 1a). The model-derived ‘uncertainty estimate’ of a noisy Bayesian model decays more slowly compared to the ‘uncertainty estimate’ of the original Bayesian model (upper panel). Second, the model-derived ‘accuracy estimate’ reflects more recent observations in a noisy Bayesian model compared to the ‘accuracy estimate’ derived from the original Bayesian model, which integrates past observations more strongly (lower panel). Hence, as mentioned beforehand, a rapid decay of uncertainty implies a small learning index; or in other words, stronger integration of past compared to recent observations.

In the following analyses, we tested whether an increasingly noisy Bayesian model mimics behaviour that is observed in the non-social compared to social condition. For example, we tested whether an increasingly noisy Bayesian model also exhibits a strongly negative ‘predictor uncertainty’ effect on interval setting (Figure 2e). In such a way, we can test whether differences in noise in the updating process of a Bayesian model might reproduce differences in learning-related behaviour seen in the social and non-social conditions.

We used these modified Bayesian models to simulate trial-wise interval setting for each participant according to the observations they made when selecting a particular advisor or non-social cue. We simulated interval setting at each trial and examined whether an increase in noise produced model behaviours that resembled participant behaviour patterns observed in the non-social condition as opposed to social condition. At each trial, we used the accuracy estimate (Methods, equation 6) – which represents a subjective belief about a single angular error -- to derive an interval setting for the selected predictor. To do so, we first derived the point-estimate of the belief distribution at each trial (Methods, equation 6) and multiplied it with the size of one interval step on the circle. The step size was derived by dividing the circle size by the maximum number of possible steps. Here is an example of transforming an accuracy estimate into an interval: let’s assume the belief about the angular error at the current trial is 50 (Methods, equation 6). Now, we are trying to transform this number into an interval for the current predictor on a given trial. To obtain the size of one interval step, the circle size (360 degrees) is divided by the maximum number of interval steps (40 steps; note, 20 steps on each side), which results in nine degrees that represents the size of one interval step. Next, the accuracy estimate in radians (0,87) is multiplied by the step size in radians (0,1571) resulting in an interval of 0,137 radians or 7,85 degrees. The final interval size would be 7,85.

We repeated behavioural analyses (Figure 2b,e,f) to test whether intervals derived from more noisy Bayesian models mimic intervals set by participants in the non-social condition: greater changes in interval setting across trials (Figure 3 —figure supplement 1b), a negative ‘predictor uncertainty' effect on interval setting (Figure 3 —figure supplement 1c), and a higher learning index (Figure 3d).

First, we repeated the most crucial analysis -- the linear regression analysis (Figure 2e) and hypothesized that intervals that were simulated from noisy Bayesian models would also show a greater negative ‘predictor uncertainty’ effect on interval setting. This was indeed the case: irrespective of social or non-social conditions, the addition of noise (increased weighting of the uniform distribution in each belief update) led to an increasingly negative ‘predictor uncertainty’ effect on confidence judgment (new Figure 3d). In Figure 3d, we show the regression weights (y-axis) for the ‘predictor uncertainty’ on confidence judgment with increasing noise (x-axis).

This new finding extends the results and suggests a theoretical account of the behavioural differences between social and non-social conditions. Increasing the noise of the belief update mimics behaviour that is observed in the non-social condition: an increasingly negative effect of ‘predictor uncertainty’ on confidence judgment. Noteworthily, there was no difference in the impact that the noise had in the social and non-social conditions, thereby suggesting that an increase in the Bayesian noise leads to an increasingly negative impact of ‘predictor uncertainty’ on confidence judgments irrespective of the condition. Hence, we can conclude that different degrees of uncertainty within the belief update is a reasonable explanation that can underlie the differences observed between social and nonsocial conditions.

Next, we used these simulated confidence intervals and repeated the descriptive behavioural analyses to test whether interval settings that were derived from more noisy Bayesian models mimic behavioural patterns observed in non-social compared to social conditions. For example, more noise in the belief update should lead to more flexible integration of new information and hence should potentially lead to a greater change of confidence judgments across predictor encounters (Figure 2b). Further, a greater reliance on recent information should lead to prediction errors more strongly in the next confidence judgment; hence, it should result in a higher learning index (Figure 2f). We used the simulated confidence interval from Bayesian models on a continuum of noise integration (i.e. different weighting of the uniform distribution into the belief update) and derived again both absolute confidence change and learning indices (Figure 3 —figure supplement 1b-c).

‘Absolute confidence change’ and ‘learning index’ increase with increasing noise weight, thereby mimicking the difference between social and non-social conditions. Further, these analyses demonstrate the tight relationship between descriptive analyses and model-based analyses thereby reinforcing both type of analyses.

We thank the reviewer for making this point, as we believe that these additional analyses allow theoretical inferences to be made in a more direct manner; we think that it has significantly contributed towards a deeper understanding of the mechanisms involved in the social and non-social conditions. Further, it provides a novel account of how we make judgments when being presented with social and non-social information.

We made substantial changes to the main text, figures and supplementary material to include these changes:

Main text, page 10-11 new section:

“The impact of noise in belief updating in social and non-social conditions

So far, we have shown that, in comparison to non-social predictors, participants changed their interval settings about social advisors less drastically across time, relied on observations made further in the past, and were less impacted by their subjective uncertainty when they did so (Figure 2). […] Correspondingly, in the non-social condition, participants express more uncertainty in their confidence estimate when they set the interval, and they also change their beliefs more quickly. Therefore, noisy Bayesian updating can account for key differences between social and non-social condition.”

Methods, page 23 new section:

“Extension of Bayesian model with varying amounts of noise

We modified the original Bayesian model (Figure 2d, Figure 2 —figure supplement 2) to test whether the integration of new evidence differed between social and non-social conditions; for example, recent observations might be weighted more strongly for non-social cues while past observations might be weighted more strongly for social cues. We tested these ideas using established ways of changing the speed of learning during Bayesian updates^14,21^. For example, an established way of changing the speed of learning in a Bayesian model is to introduce noise during the update process^14^. This noise is equivalent to adding in some of the initial prior distribution and this will make the Bayesian updates more flexible to adapt to changing environments. We added a uniform distribution to each belief update; we refer to this as noise addition to the Bayesian model. We varied the amount of noise between δ = [0,1], while δ = 0 equals the original Bayesian model and δ = 1 represents a very noisy Bayesian model. The uniform distribution was selected to match the first prior belief before any observation was made (equation 2). This δ range resulted in a continuous increase of subjective uncertainty around the belief about the angular error (Figure 3b-c).

The modified posterior distribution denoted as *p*′(σ^| x) was derived at each trial as follows:

*p*′(σ^| x) = *p*(σ^| x) + (*U*(1,140) ∗ δ)

(10)

The noisy posterior distribution as *p*′(σ^| x) was normalised as follows:

p′(σ^|x)=p′(σ^|x)∑p′(σ^|x)

We applied each noisy Bayesian model to participants’ interval settings in the social and non-social conditions. Then, we used the model to simulate interval setting at each trial with varying noise levels and then examined whether an increase in noise results in simulation behaviour that resembles behavioural patterns observed in the non-social conditions that were different to behavioural patterns observed in the social condition. At each trial, we used the accuracy estimate (equation 6) to derive a confidence interval associated with the selected predictor. We transformed the accuracy estimate – which represents a subjective belief about a single angular error – into an interval setting by multiplying the trial specific angular error with the step size. The step size was derived by dividing the circle size by the maximum number of possible steps. Here is an example of transforming an accuracy estimate into an interval: let’s assume the belief about the angular error at the current trial is 50 (Methods, equation 6). Now, we are trying to transform this number into an interval for the current predictor on a given trial. To obtain the size of one interval step, the circle size (360 degrees) is divided by the maximum number of interval steps (40 steps; note, 20 steps on each side), which results in nine degrees that represents the size of one interval step. Next, the accuracy estimate in radians (0,87) is multiplied by the step size in radians (0,1571) resulting in an interval of 0,137 radians or 7,85 degrees. The final interval size would be 7,85.”

We repeated behavioural analyses (Figure 2b,e,f) to test whether confidence intervals derived from more noisy Bayesian models mimic behavioural patterns observed in the nonsocial condition: greater changes of confidence across trials (Figure 3 —figure supplement 1b), a greater negative ‘predictor uncertainty' on confidence judgment (Figure 3 —figure supplement 1c) and a greater learning index (Figure 3d).”

Discussion, page 14:

“It may be because we make just such assumptions that past observations are used to predict performance levels that people are likely to exhibit next ^15,16^. An alternative explanation might be that participants experience a steeper decline of subjective uncertainty in their beliefs about the accuracy of social advice, resulting in a narrower prior distribution, during the next encounter with the same advisor. We used a series of simulations to investigate how uncertainty about beliefs changed from trial to trial and showed that belief updates about non-social cues were consistent with a noisier update process that diminished the impact of experiences over the longer term. From a Bayesian perspective, greater certainty about the value of advice means that contradictory evidence will need to be stronger to alter one’s beliefs. In the absence of such evidence, a Bayesian agent is more likely to repeat previous judgments. Just as in a confirmation bias ^17^, such a perspective suggests that once we are more certain about others’ features, for example, their character traits, we are less likely to change our opinions about them.”

Regarding point 2: It would be informative to pair the multivariate analysis with a whole-brain searchlight, to test whether any other regions show this effect.

We have complied with the request and conducted a whole-brain searchlight analysis. To implement the searchlight analysis, we used SPM12, RSA toolbox^22^ and custom based scripts while closely following analysis steps of previously reported whole-brain searchlight analyses^23^. We conducted the whole-brain analysis in 16 participants, because of technical difficulties matching the data from FSL to SPM. In accordance with previous studies^23^ and best-practices the searchlight analysis was based on smoothed data and multivariate noise normalization was applied to the voxel activity pattern to improve reliability^22^. The searchlight sphere was 15mm (approximately 100 voxels).

We calculated the Euclidian distance between all conditions. Next, for each condition, we calculated the Exemplar Discriminability Index (EDI) by subtracting the average Euclidian distances between neural activation maps of the same predictors from the average distances between bundles of different predictors (while taking care not to include the distance between two instances of precisely the same bundle). The resulting whole-brain maps were smoothed at 4mm.

Comparison analyses between social and non-social conditions were run on a second-level ([-1 1] for social > non-social and [1 -1] for social < non-social) to reveal the areas that responded specifically to one condition. Note the sign of each of the contrasts: we were interested in negative activation patterns, as a negative EDI is expected when

representations of the same cues are less dissimilar than representations of different cues (Figure 4g for a schematic explanation of EDI). However, we did not find a whole-brain cluster corrected result. We report this in a new supplementary figure and show subthreshold activation maps. Strong signals are apparent near our a priori ROI. Note that a comparison between the previous ROI-based analysis and the whole-brain searchlight analysis cannot be directly made and that the latter should rather be treated as a complementary analysis^23^. This is for the following reasons. The searchlight sphere comprises 100 voxels. Its spherical shape does not respect precise anatomical boundaries such as for example our pTPJ mask. Note that the dmPFC and pTPJ masks contained on average nine times the number of voxels relative to the searchlight sphere. Therefore, as in previous work (e.g., Bang et al., 2020, Nat Commun), our whole brain searchlight results are well in line with our ROI results, but they differ in degree, because of the size and shape of the neural masks used to assess neural similarity.

In addition to including this new Figure 4 —figure supplement 3, we have made corresponding changes to the main text.

Main text, section: dmPFC and pTPJ covary with social confidence and encode social advisor identity

“An exploratory whole-brain searchlight analysis was performed to test the specificity of dmPFC and pTPJ encoding the identity of social compared to non-social cues. No other cluster-corrected effects were found (Figure 4 —figure supplement 3). Additional control analyses show that neural differences between social and non-social conditions are not due to the visually different set of stimuli used in the experiment but are instead representing fundamental differences in processing social compared to non-social information (Figure 4 —figure supplement 4).**”**

References

1. Heyes, C. (2012). What’s social about social learning? Journal of Comparative Psychology 126, 193–202. 10.1037/a0025180.

2. Chang, S.W.C., and Dal Monte, O. (2018). Shining Light on Social Learning Circuits. Trends in Cognitive Sciences 22, 673–675. 10.1016/j.tics.2018.05.002.

3. Diaconescu, A.O., Mathys, C., Weber, L.A.E., Kasper, L., Mauer, J., and Stephan, K.E. (2017). Hierarchical prediction errors in midbrain and septum during social learning. Soc Cogn Affect Neurosci 12, 618–634. 10.1093/scan/nsw171.

4. Frith, C., and Frith, U. (2010). Learning from Others: Introduction to the Special Review Series on Social Neuroscience. Neuron 65, 739–743. 10.1016/j.neuron.2010.03.015.

5. Frith, C.D., and Frith, U. (2012). Mechanisms of Social Cognition. Annu. Rev. Psychol. 63, 287–313. 10.1146/annurev-psych-120710-100449.

6. Grabenhorst, F., and Schultz, W. (2021). Functions of primate amygdala neurons in economic decisions and social decision simulation. Behavioural Brain Research 409, 113318. 10.1016/j.bbr.2021.113318.

7. Lockwood, P.L., Apps, M.A.J., and Chang, S.W.C. (2020). Is There a ‘Social’ Brain? Implementations and Algorithms. Trends in Cognitive Sciences, S1364661320301686. 10.1016/j.tics.2020.06.011.

8. Soutschek, A., Ruff, C.C., Strombach, T., Kalenscher, T., and Tobler, P.N. (2016). Brain stimulation reveals crucial role of overcoming self-centeredness in self-control. Sci. Adv. 2, e1600992. 10.1126/sciadv.1600992.

9. Wittmann, M.K., Lockwood, P.L., and Rushworth, M.F.S. (2018). Neural Mechanisms of Social Cognition in Primates. Annu. Rev. Neurosci. 41, 99–118. 10.1146/annurev-neuro080317-061450.

10. Shafto, P., Goodman, N.D., and Frank, M.C. (2012). Learning From Others: The Consequences of Psychological Reasoning for Human Learning. Perspect Psychol Sci 7, 341– 351. 10.1177/1745691612448481.

11. McGuire, J.T., Nassar, M.R., Gold, J.I., and Kable, J.W. (2014). Functionally Dissociable Influences on Learning Rate in a Dynamic Environment. Neuron 84, 870–881. 10.1016/j.neuron.2014.10.013.

12. Behrens, T.E.J., Woolrich, M.W., Walton, M.E., and Rushworth, M.F.S. (2007). Learning the value of information in an uncertain world. Nature Neuroscience 10, 1214– 1221. 10.1038/nn1954.

13. Meder, D., Kolling, N., Verhagen, L., Wittmann, M.K., Scholl, J., Madsen, K.H., Hulme, O.J., Behrens, T.E.J., and Rushworth, M.F.S. (2017). Simultaneous representation of a spectrum of dynamically changing value estimates during decision making. Nat Commun 8, 1942. 10.1038/s41467-017-02169-w.

14. Allenmark, F., Müller, H.J., and Shi, Z. (2018). Inter-trial effects in visual pop-out search: Factorial comparison of Bayesian updating models. PLoS Comput Biol 14, e1006328. 10.1371/journal.pcbi.1006328.

15. Wittmann, M., Trudel, N., Trier, H.A., Klein-Flügge, M., Sel, A., Verhagen, L., and Rushworth, M.F.S. (2021). Causal manipulation of self-other mergence in the dorsomedial prefrontal cortex. Neuron.

16. Wittmann, M.K., Kolling, N., Faber, N.S., Scholl, J., Nelissen, N., and Rushworth, M.F.S. (2016). Self-Other Mergence in the Frontal Cortex during Cooperation and Competition. Neuron 91, 482–493. 10.1016/j.neuron.2016.06.022.

17. Kappes, A., Harvey, A.H., Lohrenz, T., Montague, P.R., and Sharot, T. (2020). Confirmation bias in the utilization of others’ opinion strength. Nat Neurosci 23, 130–137. 10.1038/s41593-019-0549-2.

18. Trudel, N., Scholl, J., Klein-Flügge, M.C., Fouragnan, E., Tankelevitch, L., Wittmann, M.K., and Rushworth, M.F.S. (2021). Polarity of uncertainty representation during exploration and exploitation in ventromedial prefrontal cortex. Nat Hum Behav. 10.1038/s41562-020-0929-3.

19. Yu, Z., Guindani, M., Grieco, S.F., Chen, L., Holmes, T.C., and Xu, X. (2022). Beyond t test and ANOVA: applications of mixed-effects models for more rigorous statistical analysis in neuroscience research. Neuron 110, 21–35. 10.1016/j.neuron.2021.10.030.

20. Mars, R.B., Jbabdi, S., Sallet, J., O’Reilly, J.X., Croxson, P.L., Olivier, E., Noonan, M.P., Bergmann, C., Mitchell, A.S., Baxter, M.G., et al. (2011). Diffusion-Weighted Imaging Tractography-Based Parcellation of the Human Parietal Cortex and Comparison with Human and Macaque Resting-State Functional Connectivity. Journal of Neuroscience 31, 4087– 4100. 10.1523/JNEUROSCI.5102-10.2011.

21. Yu, A.J., and Cohen, J.D. Sequential effects: Superstition or rational behavior? 8.

22. Nili, H., Wingfield, C., Walther, A., Su, L., Marslen-Wilson, W., and Kriegeskorte, N. (2014). A Toolbox for Representational Similarity Analysis. PLoS Comput Biol 10, e1003553. 10.1371/journal.pcbi.1003553.

23. Lockwood, P.L., Wittmann, M.K., Nili, H., Matsumoto-Ryan, M., Abdurahman, A., Cutler, J., Husain, M., and Apps, M.A.J. (2022). Distinct neural representations for prosocial and self-benefiting effort. Current Biology 32, 4172-4185.e7. 10.1016/j.cub.2022.08.010.

[Editors’ note: further revisions were suggested prior to acceptance, as described below.]

Reviewer #2 (Recommendations for the authors):I thank the authors for their thoughtful and thorough responses to reviewer comments. My past review raised two key concerns. First, I suggested that the paper could engage more deeply with past work on the computational basis of social learning, in order to ground their discussion on the differences between social and non-social learning. Second, I raised the concern that condition differences between social and non-social stimuli could be driven by lower-level features, such as the attentional salience or visual distinctiveness of the stimuli. Overall, I believe that the authors have gone above and beyond to address these concerns.

We thank the reviewer for these kind words and appreciate that our effort to address all comments are acknowledged. We believe that the suggested changes that were raised by the reviewer have improved the manuscript – now the findings are better embedded in previous literature, while they are also additionally extended with a theoretical Bayesian model.

First, the authors revised their introduction and discussion to cite past scholarship in social learning. But, beyond that, they also substantially expanded their analyses to disentangle differences between the underlying mechanisms driving behavior in social and non-social conditions. The authors found that participants did not have different prior expectations about the performance of social vs. non-social predictors. Instead, using a modified, noisy Bayesian model, the authors suggest that differences between conditions may be driven by degrees of uncertainty in the belief update and that greater noise in the update process leads to larger changes in confidence across different encounters with the predictor. I agree that these analyses have significantly improved the manuscript and made a new theoretical contribution in their own right, as they suggest a common computational mechanism underlying observed behavioral differences between conditions.

We thank the reviewer again for acknowledging our work and the effort we have put into the revision. We are happy to hear that the reviewer also believes that these additional analyses have extended our prior results.

If I can offer a small suggestion: Because these analyses and hypotheses were not planned in the initial submission, it would be helpful to specify that they are exploratory/an extension when they are first presented in the main text (p. 10-11).

We have made the following changes to the paragraph related to the Bayesian analyses, as well as to the discussion:

Paragraph: ‘The impact of noise in belief updating in social and non-social conditions’, page 10:

“In further exploratory analyses, we used Bayesian simulation analyses to investigate whether a common mechanism might underlie these behavioural differences. We tested whether the integration of new evidence differed between social and non-social conditions; for example, *recent* observations might be weighted more strongly for non-social cues while *past* observations might be weighted more strongly for social cues.”

Discussion, page 15:

“An alternative explanation might be that participants experience a steeper decline of subjective uncertainty in their beliefs about the accuracy of social advice, resulting in a narrower prior distribution, during the next encounter with the same advisor. During a series of exploratory analyses, we used Bayesian simulations to investigate how uncertainty about beliefs changed from trial to trial and showed that belief updates about non-social cues were consistent with a noisier update process that diminished the impact of experiences over the longer term. From a Bayesian perspective, greater certainty about the value of advice means that contradictory evidence will need to be stronger to alter one’s beliefs.”

Second, the authors added several control analyses as a supplement. In order to test whether condition differences could be accounted for by attentional engagement, the authors inspected the constant of the GLM. They found no condition differences unaccounted for by the parametric regressors included in the GLM, either in attentional regions or in the regions that were the focus of their study. Next, in order to test whether social and non-social stimuli differed in their visual distinctiveness, the authors used RSA to compare the EDI between conditions in early visual areas. Again, the authors found no difference, suggesting that observed condition differences are unlikely to be driven by these lower-level features.

Thank you for providing such a comprehensive summary, we appreciate the effort that went into reviewing our manuscript.